# Physics-Informed Deep Inverse Operator Networks for Solving PDE Inverse Problems

**Sung Woong Cho**
Stochastic Analysis and Application Research Center
Korea Advanced Institute of Science and Technology
Daejeon, South Korea
swcho95kr@kaist.ac.kr

**Hwijae Son**[*]
Department of Mathematics
Konkuk University
Seoul, South Korea
hwijaeson@konkuk.ac.kr

## ABSTRACT

Inverse problems involving partial differential equations (PDEs) can be seen as discovering a mapping from measurement data to unknown quantities, often framed within an operator learning approach. However, existing methods typically rely on large amounts of labeled training data, which is impractical for most real-world applications. Moreover, these supervised models may fail to capture the underlying physical principles accurately. To address these limitations, we propose a novel architecture called Physics-Informed Deep Inverse Operator Networks (PI-DIONs), which can learn the solution operator of PDE-based inverse problems without labeled training data. We extend the stability estimates established in the inverse problem literature to the operator learning framework, thereby providing a robust theoretical foundation for our method. These estimates guarantee that the proposed model, trained on a finite sample and grid, generalizes effectively across the entire domain and function space. Extensive experiments are conducted to demonstrate that PI-DIONs can effectively and accurately learn the solution operators of the inverse problems without the need for labeled data.

## 1 INTRODUCTION

Deep learning has revolutionized numerous fields, from natural language processing to computer vision, due to its ability to model complex patterns in large datasets (LeCun et al., 2015). In the domain of scientific computing, deep learning offers a promising approach to solving problems traditionally addressed by numerical methods, particularly when dealing with high-dimensional or nonlinear problems (Carleo et al., 2019; Karniadakis et al., 2021). However, direct applications of neural networks in scientific fields often encounter challenges such as the need for large datasets and difficulties in enforcing known physical laws within the learning process. These challenges have led to the development of specialized machine learning approaches that can integrate the governing principles of scientific problems directly into the learning framework, improving both performance and generalizability.

Physics-Informed Machine Learning (PIML) has emerged as a powerful paradigm to address these challenges by embedding physical laws, typically expressed as partial differential equations (PDEs), into the structure of neural networks (Raissi et al., 2019; Sirignano & Spiliopoulos, 2018). Rather than solely relying on large labeled datasets, PIML incorporates the governing equations of physical systems into the learning process, allowing for data-efficient models that respect the underlying physics. This paradigm is particularly useful for scenarios where the amount of available data is limited, or where it is essential to maintain the consistency of predictions with known physical principles. PIML has been applied to a wide range of problems, including fluid dynamics, heat transfer, and electromagnetics, demonstrating the ability of neural networks to capture the behavior of complex systems while obeying their physical constraints (Mao et al., 2020; Cai et al., 2021a;b; Cuomo et al., 2022).

At the early stage, PIML methods focused on solving a single problem instance, leading to a growing need for real-time inference to multiple cases. Recent advances in operator learning have aimed to

---

[*]Corresponding author.

address this by learning mappings between function spaces, significantly reducing prediction costs. The Fourier Neural Operator (FNO) and convolutional neural operator use convolution operations to learn operators on regular grids (Li et al., 2021; Raonic et al., 2023), while the multipole graph kernel network, factorized Fourier neural operator, and geometry informed neural operator are designed to handle irregular grid structures (Li et al., 2020b; Tran et al., 2023; Li et al., 2023). To predict solutions at arbitrary points, Deep Operator Network (DeepONet) (Lu et al., 2021) and HyperDeepONet (Lee et al., 2023) employ neural networks to represent output functions. Despite their efficiency, these methods require a labeled pair of functions for training. To address scenarios where such data is unavailable, physics-informed operator learning frameworks such as Physics-Informed Deep Operator Network (PI-DeepONet) and Physics-Informed Neural Operator (PINO) have been introduced (Li et al., 2024; Wang et al., 2021b), incorporating physical laws into the learning process to ensure accurate solutions even in the absence of labeled data.

Inverse problems, which aim to infer unknown parameters or functions from observed data, are a crucial area in scientific computing (Groetsch & Groetsch, 1993; Tarantola, 2005). These problems often arise in medical imaging, geophysics, and structural engineering. While traditional methods for solving inverse problems rely on optimization techniques or iterative solvers, recent advancements in deep learning have led to the development of neural network approaches for inverse problems (Aggarwal et al., 2018; Ongie et al., 2020; Fan & Ying, 2020; Bar & Sochen, 2021; Pokkunuru et al., 2023; Son & Lee, 2024).

Recently, various operator learning methods have been developed to solve inverse problems by learning the mapping between data and unknown quantities (Wang & Wang, 2024; Kaltenbach et al., 2023; Li et al., 2024), with Neural Inverse Operators (NIOs) standing out as a notable approach (Molinaro et al., 2023). However, being formulated in a supervised manner, these methods require large amounts of labeled training data, which is often impractical in real-world applications, and they may overlook important physical constraints. Furthermore, while neural operators are highly flexible, they often lack the theoretical guarantees needed to ensure stable and accurate solutions, especially in ill-posed inverse problems. This paper addresses the issues mentioned above by proposing a novel operator learning framework, termed Physics-Informed Deep Inverse Operator Networks (PI-DIONs). We summarize our main contribution as follows:

- We present PI-DIONs, a novel architecture that extends physics-informed operator learning for inverse problems. This approach enables real-time inference in arbitrary resolution without needing labeled data, effectively incorporating known physical principles.
- By integrating the stability estimates of inverse problems into the operator learning framework, we provide a theoretically robust approach for learning solution operators in PDE-based inverse problems.
- Extensive numerical validation demonstrates that PI-DIONs can accurately and efficiently learn solution operators across a range of inverse problems while removing the dependence on large, labeled datasets.

## 2 PHYSICS-INFORMED DEEP INVERSE OPERATOR NETWORKS

Let $\Omega \subset \mathbb{R}^d$ be a bounded open set with a smooth boundary, and let $\Omega_T = \Omega \times [0, T]$. Throughout this paper, we denote the domain by $\Omega$ and $\Omega_T$ depending on whether the problem is time-dependent. We consider a generic PDE defined by:

$$\begin{aligned} \mathcal{N}(u, s) = 0, & \quad x \in \Omega, \\ \mathcal{B}(u) = 0, & \quad x \in \partial\Omega. \end{aligned} \tag{1}$$

A large class of PDE-based inverse problems involves identifying an inverse operator,

$$\mathcal{F}^{-1} : u\big|_{\Omega_m} \mapsto s,$$

where $\Omega_m \subset \overline{\Omega}$ represents the measurement domain and $s$ is an unknown quantity within the system of interest. Recently, research has focused on directly parameterizing the inverse operator using neural networks $\mathcal{F}_\xi^{-1}$ supervised by paired input-output dataset $\{(u^{(i)}\big|_{\Omega_m}, s^{(i)})\}_{i=1}^N$. However, from a practical perspective, such labeled pairs are often difficult to obtain. Furthermore, this type of supervised training tends to overlook important physical constraints unless explicitly addressed (Karniadakis et al., 2021; Wang et al., 2021b). Existing operator learning frameworks do not parameterize

$u$ and $s$ as functions; instead, they treat them solely as input and output under direct supervision, which prevents the incorporation of physics-informed training. This challenge motivates the development of PI-DIONs, which we propose in this paper.

PI-DIONs can be viewed as an extension of the physics-informed DeepONets consisting of three main components. The trunk network, usually modeled with a Multi-Layer Perceptron (MLP), takes batched collocation points $x$ as input and outputs a discretized basis functions $t_\theta(x) = (t_{\theta,1}(x), t_{\theta,2}(x), \ldots, t_{\theta,p}(x))$. The reconstruction branch network processes the partial measurement data $u\big|_{\Omega_m}$ to produce the coefficients $b_\eta = (b_{\eta,1}, b_{\eta,2}, \ldots, b_{\eta,p})$ forming the complete profile of the solution $u_{\eta,\theta}(x) = b_\eta \cdot t_\theta(x) = \sum_{h=1}^p b_{\eta,h} t_{\theta,h}(x)$. Similarly, the inverse branch network takes the same partial measurement data $u\big|_{\Omega_m}$ as input and produces the coefficients $b_\zeta = (b_{\zeta,1}, b_{\zeta,2}, \ldots, b_{\zeta,p})$ for the target function $s_{\zeta,\theta}(x) = \sum_{h=1}^p b_{\zeta,h} t_{\theta,h}(x)$. This parameterization allows for the direct incorporation of the physics-informed training paradigm into inverse problems, as both $u_{\eta,\theta}$ and $s_{\zeta,\theta}$ are expressed as functions of $x$.

The loss function $\mathcal{L}$ for training PI-DIONs is defined as the sum of two components, $\mathcal{L}_{physics}$, and $\mathcal{L}_{data}$ as follows:

$$\mathcal{L}_{physics} = \frac{1}{NK} \sum_{\substack{i=1,k=1, \\ x_k \in \Omega}}^{N,K} \left[\mathcal{N}(u_{\eta,\theta}^{(i)}(x_k), s_{\zeta,\theta}^{(i)}(x_k))\right]^2 + \frac{1}{NM} \sum_{\substack{i=1,j=1, \\ x_j \in \partial\Omega}}^{N,M} \left[\mathcal{B}(u_{\eta,\theta}^{(i)}(x_j))\right]^2,$$

$$\mathcal{L}_{data} = \frac{1}{NL} \sum_{\substack{i=1,l=1, \\ x_l \in \Omega_m}}^{NL} \left[u_{\eta,\theta}^{(i)}(x_l) - u_l^{(i)}\right]^2,$$

$$\mathcal{L} = \lambda_1 \mathcal{L}_{physics} + \lambda_2 \mathcal{L}_{data}, \text{ for some } \lambda_1 \text{ and } \lambda_2,$$

where $N$ denotes the number of samples, $K, M$ denote the number of collocation points in $\Omega, \partial\Omega$, respectively, $L$ represents the number of measurement data points in $\Omega_m$, and $u_l^{(i)}$ represents the partial measurement at $x_l \in \Omega_m$. Since we model $u_{\eta,\theta}, s_{\zeta,\theta}$ as functions of $x$, the differential operator $\mathcal{N}$ can be computed using automatic differentiation at the collocation points. Thus, the physics loss $\mathcal{L}_{physics}$ ensures that both the solution and the target adhere to the physical principles, while the data loss $\mathcal{L}_{data}$ reduces the error between the reconstruction $u_{\eta,\theta}$ and the measurement data $u\big|_{\Omega_m}$. The overall architecture is illustrated in Figure 1. We also present an initial version, PI-DIONs-v0, which features another natural architecture for solving inverse problems but demonstrates inferior performance, in Appendix A.

## 3 EXTENDING STABILITY ESTIMATES TO THE OPERATOR LEARNING FRAMEWORK

The theoretical foundation for the proposed PI-DIONs is threefold. First, we introduce the stability estimates for the inverse problems we considered. This stability estimate is a crucial component that connects the prediction error with the loss functions used for training PI-DIONs. Second, we demonstrate that this stability estimate can be extended to the operator learning framework demonstrating that small enough $\mathcal{L}$ implies a small prediction error. Finally, we present a universal approximation theorem for PI-DIONs, which guarantees that the loss function can be reduced to an arbitrarily small value.

### 3.1 STABILITY ESTIMATES FOR INVERSE PROBLEMS

We start with a brief introduction to the stability estimates for inverse problems, beginning with an informal discussion of the connection between these stability estimates and the physics-informed loss functions.

**Informal statements.** Let $u$ be the solution to equation 1 for a given $s$, with partial measurements taken over the subdomain $\Omega_m \subset \Omega$. For any approximations $u^*$ and $s^*$, the loss functions $\mathcal{L}_{physics}$ and $\mathcal{L}_{data}$ are computed with $N = 1$ (i.e., the single-sample case) at the collocation points $\{x_k\}_{k=1}^K$,

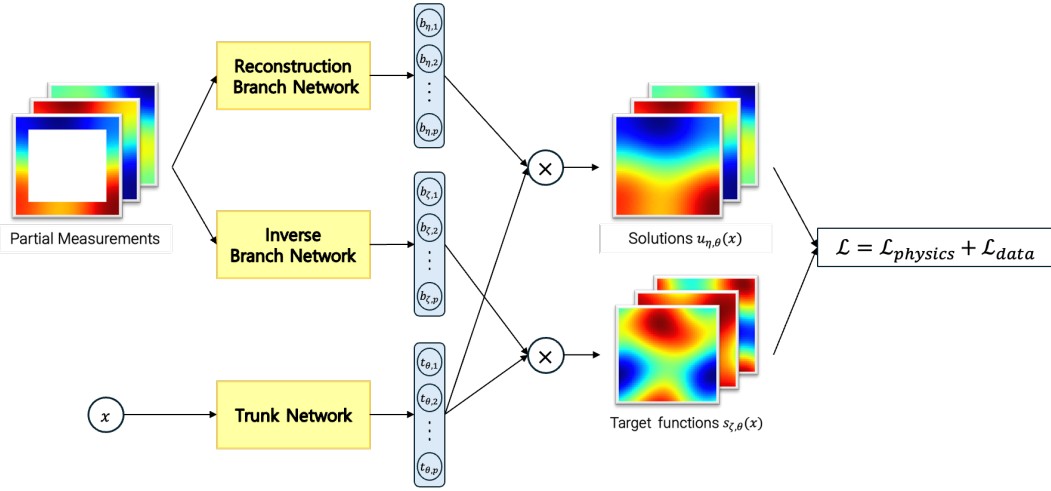

Figure 1: Schematic illustration of PI-DIONs architecture. The reconstruction and inverse branch networks take partial measurement data as inputs and produce the coefficient vector for the solution and the target function, respectively. The trunk network takes the collocation point $x$ as input and generates the corresponding basis functions for both the solution and the target function. All the networks are trained by simultaneously minimizing the loss function $\mathcal{L} = \mathcal{L}_{physics} + \mathcal{L}_{data}$.

$\{x_j\}_{j=1}^M, \{x_l\}_{l=1}^L$. The stability estimate is given by the following:

$$\|u^* - u\|_{L^2(\Omega)} + \|s^* - s\|_{L^2(\Omega)} \leq \mathcal{L}_{physics} + \mathcal{L}_{data} + \varepsilon,$$

where $\varepsilon$ converges to zero as $K$, $M$, and $L$ tend to infinity.

This stability estimate ensures that the solution $u$ and the target $s$ can be accurately approximated in the $L^2$ sense, given a sufficient number of collocation points and sufficient minimization of the loss functions. Similar estimates for inverse problems have been explored in various studies, including those derived in works such as Zhang et al. (2023) and Zhang & Liu (2023a). We now present the formal statements for the benchmark problems we considered.

**Reaction-Diffusion equation.** As a benchmark problem, we consider the reaction-diffusion equation:

$$\begin{aligned}
\partial_t u + \Delta u &= f(x)g(t), & (x,t) &\in \Omega_T := \Omega \times [0,T], \\
u(x,t) &= 0, & (x,t) &\in \partial\Omega \times [0,T], \\
u(x,0) &= u_0(x), & x &\in \Omega,
\end{aligned}$$

where $\Omega$ is a bounded domain. We assume that the function $g(t) \in L^\infty(0,T)$ is known in advance and satisfies $g^+ \geq g(t) \geq g^- > 0$ for some constants $g^+$ and $g^-$. We consider the final data $u(\cdot, T) \in H^2(\Omega)$, along with the initial condition $u_0(x)$ and the Dirichlet boundary condition, such that $\Omega_m = \partial\Omega_T$ in this example. The objective is to approximate both the solution $u$ and the source term $f$ using this boundary measurement. Let $(u^*, f^*)$ denote the approximate solution to the inverse problem. By applying regularity theory and auxiliary function techniques, the stability estimates are given as follows (See Zhang et al. (2023) for details).

$$\begin{aligned}
\|f^* - f\|_{L^2(\Omega)} &+ \|u^* - u\|_{C([0,T];L^2(\Omega))} \\
&\leq C_R(\|\partial_t u^* + \Delta u^* - f^* g\|_{H^1(0,T;L^2(\Omega))} + \|\Delta(u(x,T) - u^*(x,T)\|_{L^2(\Omega)} \\
&\quad + \|u(x,0) - u^*(x,0)\|_{L^2(\Omega)} + \|\Delta(u(x,0) - u^*(x,0))\|_{L^2(\Omega)} \\
&\quad + \|u(x,t) - u^*(x,t)\|_{H^2(0,T;L^2(\partial\Omega))}).
\end{aligned}$$

where the constant $C_R$ depends on $g^+, g^-, \Omega$, and $T$.

**Helmholtz equation.** We consider the Helmholtz equation in a bounded, connected domain $\Omega \subset \mathbb{R}^d$ with a smooth boundary $\partial\Omega$. The boundary value problem is given by:

$$\mathcal{N}u := -\nabla \cdot (\sigma \nabla u)(x) + c(x)u(x) = f(x), \quad x \in \Omega,$$

$$\mathcal{B}u := \sigma\frac{\partial u}{\partial \nu}(x) = g(x), \quad x \in \partial\Omega,$$

where $0 < \sigma_0 < \sigma(x) \in C^1(\overline{\Omega})$ and $0 < c_0 \leq c(x) \in C(\overline{\Omega})$ for some constants $\sigma_0$ and $c_0$. Here, $\nu$ represents the outward normal direction on $\partial\Omega$. The inverse source problem involves determining the unknown source function $f(x)$ by using the partial measurement of $u$ on a subdomain $\Omega_m$.

Assume that $\sigma$, $c$, and $f$ are analytic within $\Omega$, and there exist constants $\rho_1$ and $\rho_2$ such that

$$|D^\alpha f(x)| \leq \frac{\rho_1 \alpha!}{\rho_2^{|\alpha|}},$$

for all multi-indices $\alpha \in (\mathbb{N} \bigcup \{0\})^n$. According to Zhang & Liu (2023a), there exist constants $C_1$ and $C_2$ (with $C_2 \in (0,1)$) such that the following inequality holds:

$$\|f\|_{L^2(\Omega_0)} \leq C_1 \|u|_{\Omega_m}\|_{L^2(\Omega_m)}^{C_2 |\Omega_m|/|\Omega|},$$

where $\overline{\Omega}_m$ and $\overline{\Omega}_0$ are disjoint subsets of $\Omega$. For the approximate solution $(u^*, f^*)$, the stability estimates are given by:

$$\|f - f^*\|_{L^2(\Omega_0)} \leq C\big(\|u - u^*\|_{L^2(\Omega_m)}^{C_2 \frac{|\Omega_m|}{|\Omega|}} + \|\nabla \cdot (\sigma \nabla u^*) - cu^* + f^*\|_{L^2(\Omega)}^{C_2 \frac{|\Omega_m|}{|\Omega|}} + \|\sigma \frac{\partial u^*}{\partial \nu} - g\|_{L^2(\partial\Omega)}^{C_2 \frac{|\Omega_m|}{|\Omega|}}\big),$$

$$\|u - u^*\|_{H^1(\Omega)} \leq \|\nabla \cdot (\sigma \nabla u^*) - cu^* + f^*\|_{L^2(\Omega)} + \|f - f^*\|_{L^2(\Omega)} + \|\sigma \frac{\partial u^*}{\partial \nu} - g\|_{L^2(\partial\Omega)}.$$

On the other hand, the term $\|f - f^*\|_{L^2(\Omega_m)}$ can also be bounded by $\|u - u^*\|_{L^2(\Omega_m)}^{C_2 |\Omega_m|/|\Omega|}$, as previously established. By applying this bound, along with the assumption that $\|f - f^*\|_{L^2(\Omega\backslash\{\overline{\Omega}_m \bigcup \overline{\Omega}_0\})}$ is less than $\varepsilon$ and using the triangle inequality, we conclude that there exists a constant $C_H$ such that the following inequality holds.

$$\|f - f^*\|_{L^2(\Omega_0)} + \|f - f^*\|_{L^2(\Omega_m)} + \|u - u^*\|_{H^1(\Omega)}$$
$$\leq C_H\big(\|u - u^*\|_{L^2(\Omega_m)}^{C_2 \frac{|\Omega_m|}{|\Omega|}} + \|\nabla \cdot (\sigma \nabla u^*) - cu^* + f^*\|_{L^2(\Omega)}^{C_2 \frac{|\Omega_m|}{|\Omega|}} + \|\sigma \frac{\partial u^*}{\partial \nu} - g\|_{L^2(\partial\Omega)}^{C_2 \frac{|\Omega_m|}{|\Omega|}}\big) + \varepsilon.$$

### 3.2 STABILITY ESTIMATES IN THE OPERATOR LEARNING FRAMEWORK

We introduce a continuous version $\widetilde{\mathcal{L}}_{\text{data}}$ of $\mathcal{L}_{\text{data}}$ on $\Omega_m$ as follows:

$$\widetilde{\mathcal{L}}_{data} = \int_{\mathcal{U}} \int_{\Omega_m} \big[u_{\eta,\theta}(x) - u(x)\big]^2 dx d\mu(\mathcal{U}),$$

where $u_{\eta,\theta}$ is produced by the input $u|_{\Omega_m}$ and $\mu(\mathcal{U})$ represents the probability measure on the function space $\mathcal{U}$. Similarly, let $\nu(\mathcal{S})$ denote the probability measure on the function space $\mathcal{S}$ which can be induced by the inverse operator $\mathcal{F}^{-1}$ from $\mu(\mathcal{U})$. We assume the dataset $\{(u^{(i)}, s^{(i)})\}_{i=1}^N$ is sampled from $\mu(\mathcal{U})$ and $\nu(\mathcal{S})$ by using this probability measure.

We now state a theorem that implies the difference between $\widetilde{\mathcal{L}}_{data}$ and $\mathcal{L}_{data}$ becomes small when both the number of sampled functions $N$ and the number of grid points $L$ for $\mathcal{L}_{data}$ are sufficiently large.

**Theorem 1.** *Suppose that* $\sup_{u\in\mathcal{U}} \|u\|_{L^\infty(\Omega_m)} \leq R$ *and* $\sup_{\eta,\theta} \|u_{\eta,\theta}\|_{L^\infty(\Omega_m)} \leq R$ *for some* $R > 0$. *Consider an input-output dataset* $\{(u^{(i)}|_{\Omega_m}, s^{(i)})\}_{i=1}^N$ *generated through the following process. First, sample $N$ functions* $\{s^{(i)}\}_{i=1}^N$ *from the probability measure* $\nu(\mathcal{S})$ *and compute the numerical solutions* $\{u^{(i)}\}_{i=1}^N$. *Second, for all* $\{u^{(i)}\}$, *sample the grid points for partial measurements from*

the uniform distribution on $\Omega_m$ i.e., the grid points are shared across all functions. If the number of sampled functions $N$ and the number of grid points $L$ satisfy:

$$N \geq 8 \frac{\log(8N_c/\delta)}{\log 2}, \; L \geq \frac{256 R^4 |\Omega_m|^2 \log 2}{\epsilon^2},$$

then,

$$\widetilde{\mathcal{L}}_{data} \leq \mathcal{L}_{data} + \epsilon.$$

holds with probability at least $1 - \delta$, where $N_c$ is a constant depending on $\epsilon$.

The proof of this theorem leverages a symmetrization technique, analogous to the approach used in Baxter (2000). This method bounds the difference of $\mathcal{L}_{data}$ and $\widetilde{\mathcal{L}}_{data}$ by analyzing the discrepancy between two instances of $\mathcal{L}_{data}$ computed from different datasets. We can apply Hoeffding's inequality to prove that the difference converges to zero as the number of grid points and the number of sample functions increase. Further technical details of the proof are provided in Appendix D.

A similar property can be derived for $\mathcal{L}_{physics}$, where arbitrary $u, s$ sampled from $\mu(\mathcal{U})$ and $\nu(\mathcal{S})$ satisfy the governing PDE, equation 1, given a sufficiently large number of grid points and samples. As before, we define the continuous version $\widetilde{\mathcal{L}}_{physics}$ of $\mathcal{L}_{physics}$ as follows:

$$\widetilde{\mathcal{L}}_{physics} = \int_{\mathcal{U}} \int_{\Omega} \left[ \mathcal{N}(u_{\eta,\theta}(x), s_{\zeta,\theta}(x)) \right]^2 dx d\mu(\mathcal{U}) + \int_{\mathcal{U}} \int_{\Omega} \left[ \mathcal{B}(u_{\eta,\theta}(x)) \right]^2 dx d\mu(\mathcal{U}),$$

where $u_{\eta,\theta}$ and $s_{\zeta,\theta}$ are produced by the input $u\big|_{\Omega_m}$. By applying a similar technique as above, we can prove the following theorem, which implies that $\widetilde{\mathcal{L}}_{physics}$ converges to $\mathcal{L}_{physics}$ as the number of grid points $K, M$ and the number of samples $N$ becomes sufficiently large.

**Theorem 2.** *Consider the same sampling process for $u^{(i)}$ as described in Theorem 1 holds. Suppose that $\sup_{u \in \mathcal{U}} \|\mathcal{N}u\|_{L^\infty(\Omega_m)} \leq R_{\mathcal{N}}, \sup_{\eta,\theta} \|\mathcal{N}u_{\eta,\theta}\|_{L^\infty(\Omega_m)} \leq R_{\mathcal{N}}$, and $\sup_{u \in \mathcal{U}} \|\mathcal{N}u\|_{L^\infty(\Omega_m)} \leq R_{\mathcal{B}}, \sup_{\eta,\theta} \|\mathcal{N}u_{\eta,\theta}\|_{L^\infty(\Omega_m)} \leq R_{\mathcal{B}}$. Additionally, let $K$ and $M$ denote the number of grid points for the interior $\Omega$ and the boundary $\partial\Omega$, respectively, and shared across all $u^{(i)}$. If the number of sampled functions $N$ and the number of grid points $K$, $M$ satisfy:*

$$N \geq \max \left( \frac{8 \log(16 N_{\mathcal{N}}/\delta)}{\log 2}, \frac{8 \log(16 N_{\mathcal{B}}/\delta)}{\log 2} \right), \; K \geq \frac{64 R_{\mathcal{N}}^4 |\Omega|^2 \log 2}{\epsilon^2}, M \geq \frac{64 R_{\mathcal{B}}^4 |\partial\Omega|^2 \log 2}{\epsilon^2},$$

*then with probability at least $1 - \delta$, the loss functions will satisfy*

$$\widetilde{\mathcal{L}}_{physics} \leq \mathcal{L}_{physics} + \epsilon,$$

*where $N_{\mathcal{N}}$ and $N_{\mathcal{B}}$ are constants depending on $\epsilon$.*

By applying Theorems 1 and 2, we can now derive the following theorem, which states that the output functions $u_{\eta,\theta}$ and $s_{\zeta,\theta}$ of PI-DIONs can approximate the true $u$ and $s$ with high probability over $\mu(\mathcal{U})$ and $\nu(\mathcal{S})$. Notably, previous results from Zhang et al. (2023); Zhang & Liu (2023a) established a similar stability estimate, but they considered only the special case where the supports of $\mu(\mathcal{U})$ and $\nu(\mathcal{S})$ consist of a single element within $\mathcal{U}$ and $\mathcal{S}$, respectively. In contrast, this paper aims to extend the result by considering more general distributions $\mu(\mathcal{U})$ and $\nu(\mathcal{S})$. Consequently, the theorem below implies that PI-DIONs can accurately approximate $u$ and $s$ even with the partial measurement of unseen data $u$ in $\mathcal{U}$, which represents our main theoretical result for PI-DIONs.

**Theorem 3.** *Suppose the same sampling process holds as in Theorem 2. Additionally, assume that the number of sampled functions $N$ and the number of grid points $L$, $K$, $M$ satisfy the conditions in Theorem 2. Then, for any $u \in \mathcal{U}, s \in \mathcal{S}$, and $\varepsilon > 0$, the following inequality holds with probability at least $(1 - 2\delta)(1 - 2\sqrt{\epsilon} - \frac{\mathcal{L}_{physics} + \mathcal{L}_{data}}{\sqrt{\epsilon}})$,*

$$\|u_{\eta,\theta} - u\|_{L^2(\Omega)} + \|s_{\zeta,\theta} - s\|_{L^2(\Omega)} \leq \sqrt{\varepsilon}.$$

**Remark 1.** *Theorem 3 applies to the inverse source problem for the reaction-diffusion equation using boundary measurements, as well as to the inverse source problem for the Helmholtz equation using internal measurements. We believe that this result can be generalized to other inverse problems, where a stability estimate for a single instance can be derived.*

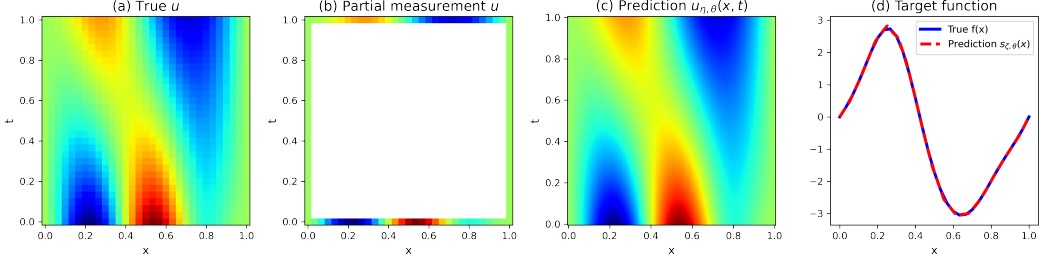

Figure 2: A test sample and results for the inverse source problem of the reaction-diffusion equation. (a) True solution $u$ on a $30 \times 30$ rectangular grid. (b) Partial measurement data $u\big|_{\partial \Omega_T}$ collected from the true solution on the same grid. (c) Predicted solution $u_{\eta,\theta}$ on a $200 \times 200$ rectangular grid. (d) The source function $f(x)$ (blue) and the predicted source function $s_{\zeta,\theta}$ (red).

### 3.3 Universal approximation theorem for PI-DIONs

Finally, we demonstrate that PI-DIONs can achieve small values for both $\mathcal{L}_{data}$ and $\mathcal{L}_{physics}$. This result builds upon the universal approximation property of DeepONet, which asserts that DeepONet can approximate certain operators, including nonlinear mappings between two compact function spaces. Specifically, we claim that if DeepONet accurately approximates the solution and inverse operators, the physics-informed loss $\mathcal{L}_{physics}$ will be small enough. The proof of this claim relies on the standard triangle inequality, under the assumption that both the solution and inverse operators are Lipschitz continuous with respect to the measurements. By combining this with the results from Theorem 3, we ensure that PI-DIONs with a small value of $\mathcal{L}$ provide accurate approximations of both $u$ and $s$.

**Proposition 1.** *(Universal approximation theorem for PI-DIONs). Assume that the branch network and trunk network in PI-DIONs have continuous, non-polynomial activation functions. For any given input-output dataset $\{(u^{(i)}\big|_{\Omega_m}, s^{(i)})\}_{i=1}^{N}$ and for any $\varepsilon > 0$ there exist appropriate parameters $\eta, \zeta,$ and $\theta$ for PI-DIONs such that*

$$\mathcal{L} = \mathcal{L}_{physics} + \mathcal{L}_{data} \leq \epsilon.$$

## 4 Experiments

We empirically evaluate the proposed PI-DIONs on three benchmark inverse problems of different types. We present both the unsupervised PI-DIONs and supervised PI-DIONs, where the supervised one is trained with an additional loss function $\mathcal{L}_s = \frac{1}{NK} \sum_{i=1,k=1}^{N,K} \left[ s_{\zeta,\theta}^{(i)}(x_k)) - s_k^{(i)} \right]^2$, where $x_k \in \Omega$ and $s_k^{(i)}$ represents the label, so that the loss function becomes

$$\mathcal{L} = \lambda_1 \mathcal{L}_{physics} + \lambda_2 (\mathcal{L}_{data} + \mathcal{L}_s), \text{ for some } \lambda_1 \text{ and } \lambda_2.$$

For the comparative analysis, we choose the supervised DeepONets and FNOs as baselines. The details of the training process, including the architecture and hyperparameters, are presented in Appendix B. Additional experiments, including a sensitivity analysis, an ablation study, and a comparison with PINNs, are provided in Appendix C. The $\lambda$-values are selected based on insights gained from training PINNs for each example. It is worth noting that across all three experiments, we observe that a larger $\frac{\lambda_2}{\lambda_1}$ results in a smaller test error. This aligns with our intuition that, during the early stages of training, a large $\mathcal{L}_{data}$ will push $s_{\zeta,\theta}$ in the wrong direction, as $u_{\eta,\theta}$ differs from the true solution, leading to an inaccurate $\mathcal{L}_{physics}$. Finally, in Appendix C.3, we provide a detailed description and additional experiments on integrating the variable-input operator network(Prasthofer et al., 2022) into PI-DIONs to address the case of irregular measurements (sensor points).

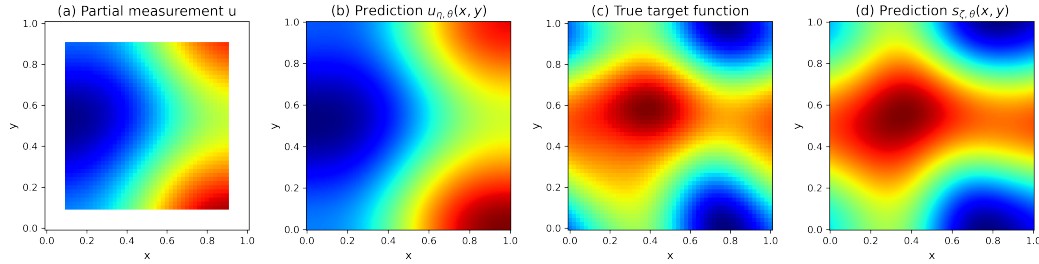

Figure 3: A test sample and results for the Helmholtz equation. (a) Partial measurement $u\big|_{\Omega_m}$ on an internal $40 \times 40$ grid. (b) Predicted solution $u_{\eta,\theta}(x,y)$ on a $200 \times 200$ rectangular grid. (c) True source function $f(x)$ on a $50 \times 50$ grid. (d) Predicted source function $s_{\zeta,\theta}(x,y)$ on a $200 \times 200$ grid.

## 4.1 REACTION-DIFFUSION EQUATION: INVERSE SOURCE PROBLEM USING BOUNDARY MEASUREMENT

We start by considering the reaction-diffusion equation

$$
\begin{aligned}
\partial_t u + \Delta u &= F(x,t), & (x,t) \in \Omega_T := \Omega \times [0,T], \\
u(x,t) &= 0, & (x,t) \in \partial\Omega \times [0,T], \\
u(x,0) &= u_0(x), & x \in \Omega,
\end{aligned}
$$

where $\Omega = [0,1]$ and $T = 1$. As it is hard to measure the internal source in many engineering applications, discovering unknown source function $F(x,t)$ from the temperature distribution $u(x,t)$ is an important inverse problem. The inverse problem amounts to discovering the source function $f(x)$ from the initial data $u(x,0)$, final data $u(x,T)$, and boundary data $u\big|_{\partial\Omega \times [0,T]}$. It is well known that if the source function $F(x,t) = f(x)g(t)$ is separable and g(t) is given in advance, then this problem attains a unique solution. Recently, Zhang et al. (2023) proposed a physics-informed neural network for this problem, but it requires retraining when new data is introduced. In contrast, we demonstrate that our method successfully learns the inverse operator, enabling real-time inference in arbitrary resolution without the need for retraining on the same problem.

We randomly sampled the initial condition $u_0(x)$ and the unknown source function $f(x)$ from the Gaussian random field. Using the central Finite Difference Method (FDM) on a $30 \times 30$ grid, we computed the numerical solution and extracted the partial measurement dataset $\{u^{(i)}\big|_{\partial\Omega_T}\}_{i=1}^N$, precisely the boundary data in the space-time domain. Since the measurement data can be treated as a 1-dimensional function, we used an MLP for both the reconstruction and inverse branch networks. As the solution and target functions have different domains, i.e., $u_{\eta,\theta} : \mathbb{R}^2 \to \mathbb{R}$ and $s_{\zeta,\theta} : \mathbb{R} \to \mathbb{R}$, we made reconstruction trunk network and inverse trunk network separately. The reconstruction branch, together with the reconstruction trunk network learns a mapping from the partial measurement $u\big|_{\partial\Omega_T}$ to the complete solution profile $u\big|_{\overline{\Omega_T}}$. Simultaneously, the inverse branch, also paired with the inverse trunk network, is trained to approximate the inverse operator $\mathcal{F}^{-1} : u\big|_{\partial\Omega_T} \mapsto f$.

We trained a PI-DION with 1000 unsupervised samples, where the data consists solely of partial measurements $u\big|_{\partial\Omega_T}$. We then compared the relative $L^2$ errors against benchmark supervised models, including DeepONets. Furthermore, we trained PI-DIONs with labeled training data $\{(u^{(i)}\big|_{\partial\Omega_T}, f^{(i)})\}$ to assess the performance of the unsupervised PI-DION. All models were evaluated on a test dataset of 1000 samples. As shown in Table 1, the unsupervised PI-DION outperforms the supervised models and achieves a test error comparable to that of PI-DION trained with fully supervised samples. It is important to note that FNO is not applicable to this problem, as it requires the input and output functions to be defined on the same domain. In this problem, input function $u\big|_{\partial\Omega_m}$ is defined on the boundary of $\partial\Omega_T$, while the source term $f$ is defined within the domain $\Omega$. This discrepancy between the domains requires training a nontrivial mapping between these two different spaces. We present a test sample of the true solution, partial measurement, predicted solution, true source function and the predicted source function in Figure 2.

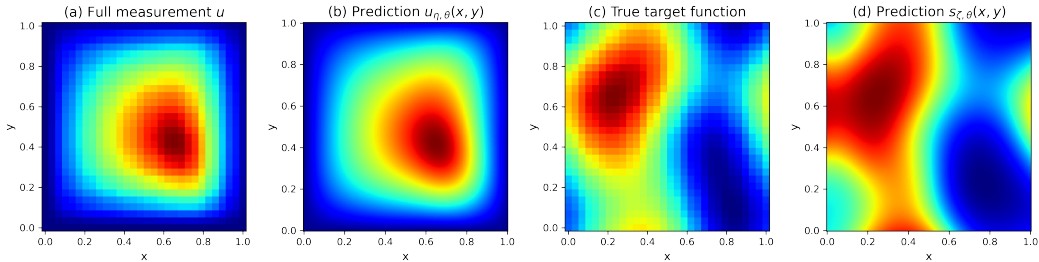

Figure 4: A test sample and results for the Darcy flow. (a) Full measurement $u$ on a $30 \times 30$ rectangular grid. (b) Predicted solution $u_{\eta,\theta}$ on a $200 \times 200$ rectangular grid. (c) True permeability field $s$ on a $30 \times 30$ grid. (d) Predicted permeability field $s_{\zeta,\theta}$ on a $200 \times 200$.

## 4.2 HELMHOLTZ EQUATION: INVERSE SOURCE PROBLEM USING INTERNAL MEASUREMENT

We consider the following Helmholtz equation:

$$-\nabla \cdot (\sigma(x)\nabla u(x)) + c(x)u(x) = f(x), \quad x \in \Omega,$$

$$\sigma(x)\frac{\partial u}{\partial \nu} = g(x), \quad x \in \partial\Omega,$$

where $\Omega = [0,1]^2$ and $\nu$ represents the outward normal direction on $\partial\Omega$. The objective of this inverse problem is to reconstruct the source function $f(x)$ for $x \in \Omega$ using internal measurement $u\big|_{\Omega_m}$, where $\Omega_m = [0.2, 0.8]^2$. For simplicity, we set $\sigma = c \equiv 1$ and $g \equiv 0$. This problem has been extensively studied from a Physics-Informed Neural Networks (PINNs) perspective in Zhang & Liu (2023b;a). Here, we extend the problem into the operator learning framework and demonstrate that PI-DION effectively solves the inverse problem by approximating the inverse operator without labeled data.

We again sampled $f(x)$ from the Gaussian random field and computed the numerical solution using FDM, to obtain the internal measurement dataset. The reconstruction branch and the trunk network learn a continuation mapping from $u\big|_{\Omega_m}$ to $u$. Likewise, the inverse branch and the trunk network learn an inverse operator $\mathcal{F}^{-1} : u\big|_{\Omega_m} \mapsto f$. Figure 3 illustrates a test sample and results. Here, we employ a CNN to model both branch networks and an MLP to model the trunk network. We trained DeepONet and FNO with 50, 500, 1000 supervised samples, while PI-DIONs were trained with 1000 samples in both supervised and unsupervised settings. Table 1 shows that supervised PI-DION achieves the lowest test error, whereas the unsupervised PI-DION yields a test error comparable to that of the supervised FNO.

## 4.3 DARCY FLOW: UNKNOWN PERMEABILITY

Next, we consider the 2D steady state Darcy flow equation:

$$-\nabla \cdot (\sigma(x)\nabla u(x)) = f(x), \quad x \in \Omega,$$

$$u(x) = 0, \quad x \in \partial\Omega.$$

In this example, $u(x)$ represents the pressure field in a porous medium, defined by a positive permeability field $\sigma(x)$. The inverse problem involves determining the unknown permeability field $\sigma(x)$ from the full measurement $u(x)$. We randomly sampled the permeability from a Gaussian random field followed by a min-max scaling, and computed the numerical solution $u$, using the fixed source term $f(x,y) = 100x(1-x)y(1-y)$ and FDM on a $30 \times 30$ grid.

In this setup, the reconstruction branch, paired with the trunk network, simply learns the identity mapping from the measurement to the solution. In contrast, the inverse branch, along with the trunk network, learns the inverse operator $\mathcal{F}^{-1} : u \mapsto \sigma$. Both branch networks are modeled using convolutional neural networks (CNN), while the trunk network is modeled using an MLP. Given that the permeability field is always positive, we impose a hard constraint on the inverse operator to ensure the output remains positive. We trained a PI-DION with 1000 unsupervised samples, where

Table 1: Relative $L^2$ error of the predicted target function $s_{\zeta,\theta}$ computed over test dataset with the best model highlighted in bold. We trained supervised PI-DIONs with $(\lambda_1, \lambda_2) = (1, 100)$ and unsupervised PI-DIONs with $(\lambda_1, \lambda_2) = (1, 100)$. Across all three benchmark problems, our PI-DION achieves the best performance, showing only a small difference between supervised and unsupervised models.

| Model | # Train data | Reaction Diffusion | Darcy Flow | Helmholtz equation |
|---|---|---|---|---|
| DeepONet | 50 | 33.60% | 20.15% | 64.17% |
| w/ MLP, CNN branch | 500 | 1.29% | 9.06% | 23.56% |
| (Supervised) | 1000 | 1.10% | 8.77% | 7.86% |
| DeepONet | 50 | 42.78% | 28.79% | 74.45% |
| w/ FNO branch | 500 | 32.23% | 12.78% | 23.61% |
| (Supervised) | 1000 | 21.10% | 9.12% | 11.30% |
| FNO | 50 | N/A | 10.11% | 38.42% |
| | 500 | N/A | 5.59% | 28.23% |
| (Supervised) | 1000 | N/A | **3.41**% | 25.85% |
| PI-DION(Ours) (Supervised) | 1000 | 1.04% | **3.45**% | **5.64**% |
| PI-DION(Ours) (Unsupervised) | 1000 | **1.03**% | 8.10% | 8.05% |

the dataset consists of $u$ only. We compared the relative $L^2$ errors against the supervised models again. The results are summarized in Table 1 and Figure 4. Supervised FNO demonstrates the best performance for this problem and the supervised PI-DION achieves a comparable test error. We believe there is room for improvement if a stability estimate along with the corresponding physics-informed loss function can be further identified for this problem.

## 5 DISCUSSION

We explored inverse problems that can be framed through the identification of an inverse operator $\mathcal{F}^{-1}$ between suitable function spaces. In this context, we proposed a novel architecture called PI-DIONs, which eliminates the reliance on costly labeled data typically required for training machine learning models. By adopting a physics-informed approach, PI-DIONs directly parameterize the functions of interest—specifically, the solution $u_{\eta,\theta}$ and the target function $s_{\zeta,\theta}$—thereby integrating the underlying physical principles into the model. Additionally, we provided a robust theoretical foundation for the proposed method by extending stability estimates for inverse problems to the operator learning framework. Our theoretical results demonstrate that PI-DIONs can accurately predict both the solution $u$ and the unknown quantity $s$ based on partial measurement data $u\big|_{\Omega_m}$. This ability to work with incomplete data highlights the practicality and effectiveness of our approach. Additionally, we believe that empirically verifying the theoretical bounds on $N, K, M$, and $L$ will be an interesting direction for future work.

To substantiate our claims, we conducted a series of numerical studies that showcased the superior performance of unsupervised PI-DIONs. Remarkably, these models achieved test errors comparable to those of traditional supervised baselines, thereby indicating that unsupervised training can be as effective as its supervised counterpart in certain contexts. While the inference time of PI-DION is significantly faster than that of a single PINN, it is important to note that the unsupervised training of PI-DION entails substantial computational costs (see Table 5). Therefore, accelerating the convergence of PI-DION presents a promising avenue for future research, potentially enabling more efficient training processes and broader applicability to complex inverse problems. Furthermore, the integration of PI-DIONs with recent advancements in DeepONets, such as those presented in Prasthofer et al. (2022) and Cho et al. (2024), will be an interesting direction for future work. This exploration will not only refine our methodology but also contribute to the broader field of physics-informed machine learning.

## 6 REPRODUCIBILITY STATEMENT

We provide detailed experimental setups for each PDE in Appendix B. Additionally, the proofs are included in Appendix D, and the source code is submitted as supplementary material.

### ACKNOWLEDGMENTS

Sung Woong Cho was supported by National Research Foundation of Korea (NRF) grant funded by the Ministry of Science and ICT (KR) (RS-2024-00462755) and funded by the Korea government (MSIT) (RS-2019-NR040050). Hwijae Son was supported by the National Research Foundation of Korea (NRF) grant funded by the Korea government (MSIT) (No. NRF-2022R1F1A1073732).

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

# A   PI-DIONs-v0

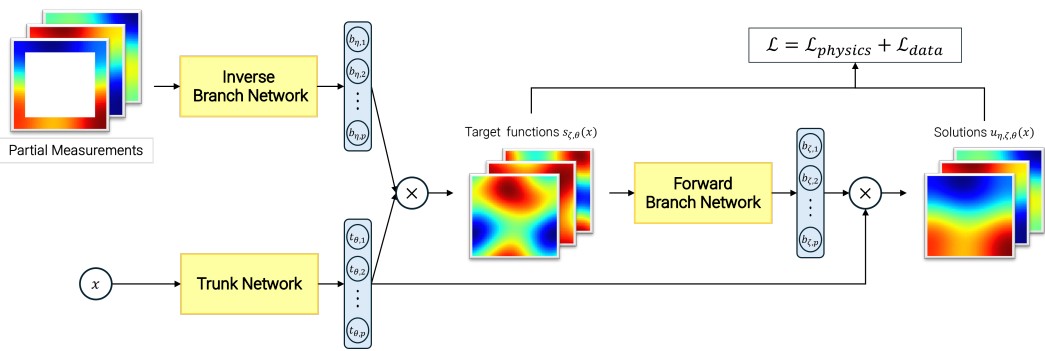

Figure 5: Schematic illustration of PI-DIONs-v0 architecture. The inverse branch network takes partial measurement data as input and produces the coefficient vector for the target function. The trunk network receives the collocation point $x$ as input and generates the corresponding basis functions for the target function. The forward branch network takes the predicted target function $s_{\zeta,\theta}$ as input and outputs the coefficients that form the solution $u_{\eta,\zeta,\theta}$. All the networks are trained by simultaneously minimizing the loss function $\mathcal{L} = \mathcal{L}_{physics} + \mathcal{L}_{data}$.

In this section, we introduce an initial version of PI-DIONs, termed PI-DION-v0, which was considered during the early stages of development. PI-DION-v0 consists of the inverse branch network, the forward branch network, and the trunk network, similar to the final PI-DIONs architecture. However, PI-DION-v0 first produces the target function solution $s_{\zeta,\theta}(x)$ by combining the outputs of the inverse branch and the trunk network, then uses this prediction as input to the forward branch network to produce the final target function $u_{\eta,\zeta,\theta}(x)$. The key difference lies in the input to the forward(reconstruction) branch: while PI-DIONs use the partial measurement, PI-DION-v0 uses the predicted target function. The overall architecture is illustrated in Figure 5.

PI-DION-v0 also achieved promising relative test errors for the inverse source problem in the reaction-diffusion equation. However, PI-DION consistently outperformed PI-DION-v0 in both supervised and unsupervised settings (see Table 2). Moreover, PI-DION proved to be computationally more efficient than PI-DION-v0. Despite this, we believe it is worth discussing the architecture of PI-DION-v0 in this paper, as it represents a more natural approach to solving the inverse problem. The first part, consisting of the inverse branch and trunk network, can be viewed as an inverse operator $\mathcal{F}^{-1}$, while the second part, the forward branch paired with the trunk network, acts as a forward operator $\mathcal{F}$. Thus, PI-DION-v0 can be interpreted as learning an identity operator $\mathcal{I} = \mathcal{F}^{-1} \circ \mathcal{F}$, mapping partial measurements to the solution. We believe there are situations where PI-DION-v0 outperforms, primarily due to its more natural architecture.

Table 2: Relative $L^2$ test errors.

| # Train data | PI-DIONs-v0 | PI-DIONs |
|---|---|---|
| Supervised w/ 1000 | 4.10% | **1.04**% |
| Unsupervised w/ 1000 | 5.62% | **1.03**% |

# B   ARCHITECTURE AND TRAINING DETAILS

All experiments were conducted on a single RTX 3090 GPU, with the batch size determined based on available memory. For the three experiments, we used either 1,000 or 500 samples per batch. We used Adam optimizer for training, with a learning rate 1e-3. We trained PI-DIONs and the baseline models for a sufficient number of epochs until convergence was achieved, as is typical with physics-informed training, which tends to have a slow yet reliable convergence process. We summarize the number of trainable parameters in Table 3.

Table 3: Number of trainable parameters for each model.

| Model | Reaction Diffusion | Darcy Flow | Helmholtz equation |
|---|---|---|---|
| DeepONet | 6K | 70K | 72K |
| DeepONet w/ FNO branch | 45K | 100K | 150K |
| FNO | N/A | 100K | 150K |
| PI-DION | 12K | 100K | 110K |

## B.1 REACTION-DIFFUSION EQUATION: INVERSE SOURCE PROBLEM USING BOUNDARY MEASUREMENT.

Both the reconstruction and inverse branch networks were simple Multi-Layer Perceptrons (MLP) each consisting of layers with 60 (input), 32, 32, and 32 (output) A ReLU activation was applied to each layer. Since the solution and target functions have different domains, i.e., $u_{\eta,\theta} : \mathbb{R}^2 \to \mathbb{R}$ and $s_{\zeta,\theta} : \mathbb{R} \to \mathbb{R}$, we designed separate trunk networks for reconstruction and inversion. Both trunk networks were also MLPs, with the reconstruction trunk consisting of layers with 2 (input), 32, 32, and 32 (output) nodes, and the inverse trunk consisting of layers with 1 (input), 32, 32, and 32 (output) nodes where a Tanh activation function was applied. The baseline DeepONet consists of the same MLP branch and MLP trunk networks and we did not consider the baseline FNO for this example.

The numerical solutions were computed on a $30 \times 30$ grid, and the same grid was used as input to the trunk networks for training PI-DIONs. The model was trained in a full-batch setting, where the input to the branch networks has a size of (1000, 60), the input to the reconstruction trunk is (900, 2), and the input to the inverse trunk is (30, 1).

## B.2 HELMHOLTZ EQUATION: INVERSE SOURCE PROBLEM USING INTERNAL MEASUREMENT

For the Helmholtz equation, we employed a convolutional neural network (CNN) composed of three Conv2d layers with $3 \times 3$ filters, each followed by the GeLU activation and max-pooling layers with a $3 \times 3$ kernel. Additionally, an MLP with 128 output nodes and GeLU activation was applied for both the reconstruction and inverse branch networks. The trunk network was an MLP with a structure consisting of 2 (input), 128, 128, and 128 (output) nodes where a Tanh activation was applied. The baseline DeepONet consists of the same CNN branch and MLP trunk networks and we adopted the implementation of FNO from Li et al. (2020a).

The numerical solutions were computed on a $50 \times 50$ grid, and the internal $40 \times 40$ grid was used as input for the branch networks. PI-DION was trained with a mini-batch size of 500, meaning the input to the branch networks had a size of (500, 1600), and the input to the trunk network had a size of (2500, 2).

## B.3 DARCY FLOW: UNKNOWN PERMEABILITY

For the Darcy flow, we employed a CNN and MLP with the same architecture as used for the Helmholtz equation. The trunk network is an MLP consisting of layers with 2 (input), 128, 128, and 128 (output) nodes. To ensure the predicted permeability lies within the range $[0, 1]$, we applied a sigmoid activation function to the prediction $s_{\zeta,\theta}$. The baseline DeepONet consists of the same CNN branch and MLP trunk networks and we adopted the implementation of FNO from Li et al. (2020a).

The numerical solutions were computed on a $100 \times 100$ grid and then downsampled to a $30 \times 30$ scale. PI-DION was trained with a mini-batch of size 500, meaning the input to the branch networks had a size of (500, 900) and the input to the trunk network had a size of (900, 2).

## C  ADDITIONAL EXPERIMENTS

In this section, we present additional experiments to further demonstrate the robustness and effectiveness of PI-DIONs.

### C.1  COMPARISON TO PHYSICS-INFORMED NEURAL NETWORKS(PINNS)

We select the weights $\lambda_1$ and $\lambda_2$ for PI-DIONs based on insights gained from training PINNs. The inverse PINNs discussed in this section are unsupervised, meaning no supervision is provided for the target function $s$. For each problem, we utilize two neural networks, each comprising three hidden layers with 64 neurons per layer, to approximate the solution $u$ and the target function $s$. The loss functions are defined similarly to those in PI-DIONs:

$$\mathcal{L} = \lambda_1 \mathcal{L}_{physics} + \lambda_2 \mathcal{L}_{data}.$$

All experiments were conducted on a single RTX 3090 GPU. We used Adam optimizer for training, with a learning rate 1e-3.

Table 4 presents the relative $L^2$ errors for different combinations of $\lambda$. Interestingly, the unsupervised PI-DIONs achieve error levels comparable to those of a single PINN, despite the significantly shorter inference time of PI-DIONs compared to the longer training time required for PINNs (see Table 5).

Table 4: Relative $L^2$ error of PINNs for different combinations of $\lambda$.

| $(\lambda_1, \lambda_2)$ | Reaction Diffusion | Darcy Flow | Helmholtz equation |
|:---:|:---:|:---:|:---:|
| (1, 1) | **0.90%** | 2.51% | 8.44% |
| (1,100) | 3.52% | **1.47%** | **7.72%** |
| (100,1) | 7.55% | 10.03% | 11.49% |

Table 5: Approximate training and inference time for PINNs and PI-DIONs.

| | | Reaction Diffusion (1e+6 epochs) | Darcy Flow (1e+7 epochs) | Helmholtz equation (1e+7 epochs) |
|:---:|:---:|:---:|:---:|:---:|
| PINNs | Training | 20m | 3h | 4h |
| PI-DIONs | Training | 5h | 24h | 24h |
| | Inference | **2ms** | **5ms** | **5ms** |

### C.2  SENSITIVITY ANALYSIS

The loss function comprising $\mathcal{L}_{physics}$ and $\mathcal{L}_{data}$ is widely used in the Physics-Informed Machine Learning(PIML) literature. Additionally, numerous studies have introduced loss-balancing algorithms for these two components (e.g., (Wang et al., 2021a; Son et al., 2023)) by multiplying weights $\lambda_1$ and $\lambda_2$ to construct an objective function:

$$\mathcal{L} = \lambda_1 \mathcal{L}_{physics} + \lambda_2 \mathcal{L}_{data}.$$

Here, we investigate how the performance of the proposed PI-DIONs can be further improved by incorporating such techniques. Specifically, we conduct a sensitivity analysis on the $\lambda$-values. For this analysis, we train seven PI-DION models with the following weight combinations:

$$(\lambda_1, \lambda_2) = (100, 1), (10, 1), (1, 1), (1, 0.1), (0.1, 1), (1, 10), (1, 100).$$

The experiments are conducted to solve the inverse source problem for the reaction-diffusion equation described in Section 4.1. Table 6 presents the relative $L^2$ errors for each model with different combinations of $(\lambda_1, \lambda_2)$. Although the single PINN for the reaction-diffusion problem achieves

Table 6: Relative $L^2$ error for different combinations of $\lambda$.

| $(\lambda_1, \lambda_2)$ | (100,1) | (10,1) | (1,1) | (1,0.1) | (0.1,1) | (1,10) | (1,100) |
|---|---|---|---|---|---|---|---|
| Relative $L^2$ error | 27.28% | 14.84% | 10.17% | 13.81% | 3.12% | 4.79% | **1.03%** |

the best error with $(\lambda_1, \lambda_2) = (1, 1)$, we observe that PI-DION achieves the best error when $(\lambda_1, \lambda_2) = (1, 100)$.

We additionally conduct an ablation study to examine the effect of sample size on the reaction-diffusion equation described in Section 4.1. Table 7 shows that the relative error decreases as the number of training samples ($N$) increases, eventually saturating around $N = 1000$.

Table 7: Relative $L^2$ error for different number of training samples.

| N | 100 | 500 | 1000 | 2000 |
|---|---|---|---|---|
| Relative $L^2$ error | 28.73% | 5.07% | 1.03% | **0.98%** |

## C.3 VARIABLE-INPUT PI-DIONS

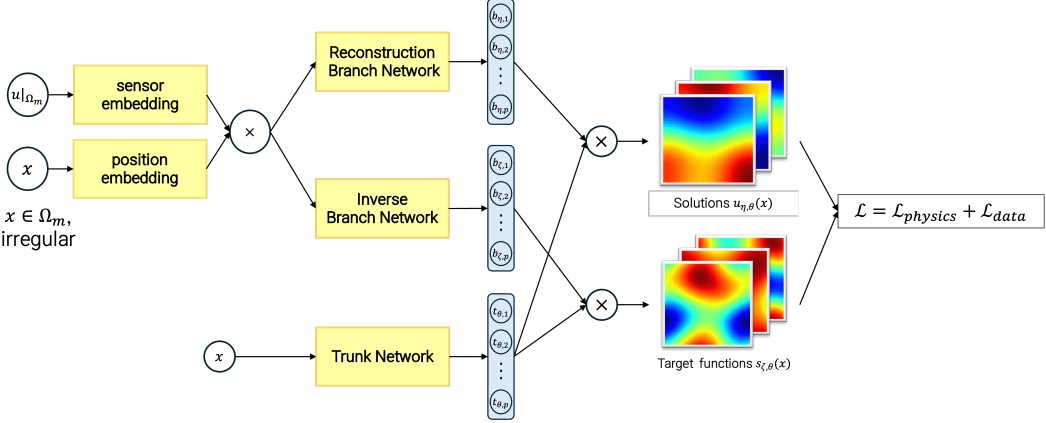

Figure 6: Schematic illustration of variable-input PI-DIONs architecture.

Vanilla DeepONet requires that the sensor points (locations where the input function is sampled) remain consistent across all samples. However, in the context of inverse problems, this constraint can be problematic, as sensor points may vary from sample to sample. To address this limitation, Prasthofer et al. (2022) introduced variable-input deep operator networks, which allow for flexibility in sensor point locations. This variable-input architecture can be seamlessly incorporated into our PI-DION framework. To evaluate this approach, we conducted experiments on a reaction-diffusion equation. We generated a dataset $\{(u^{(i)}\big|_{\Omega_m}, s^{(i)})\}_{i=1}^{N}$ using a fine computational grid $(\{(t_k, x_l)\}_{k=1,l=1}^{1000,100}$, considering the CFL condition De Moura & Kubrusly (2013)). For each sample $(u^{(i)}\big|_{\Omega_m}, s^{(i)})$, we randomly selected 30 collocation points in the spatial domain, resulting in irregularly sampled data. Consequently, the set of collocation points varies across samples.

Although Prasthofer et al. (2022) originally proposed an attention-based mechanism for handling variable input, we opted for a simplified architecture. The overall structure is depicted in Figure 6. Both the sensor embedding and position embedding are implemented using simple multilayer perceptrons (MLPs). The final embedding is obtained by computing the inner product of their outputs. For the weights, we used $(\lambda_1, \lambda_2) = (1, 100)$. In this experiment, we obtained the $L^2$ relative error of **3.83%** which is compatible with the original PI-DION results. This demonstrates that the proposed method can be effectively generalized to cases with irregular measurement points.

# D PROOF OF THEOREMS

## D.1 PROOF OF THEOREM 1

Throughout this section, we assume the existence of a sufficiently large constant $R > 0$ such that all functions $u(x) \in \mathcal{U}$ are contained within the range $[-R, R]$. Additionally, we assume that $u_{\theta,\eta}(x)$ is also contained within $[-R, R]$ for every $\theta, \eta$ and $x \in \Omega$. This condition can be ensured by using bounded parameters $\theta, \eta$ particularly when the activation function of the trunk network is Lipschitz continuous. Specifically, $u(x_l)$ is bounded (by the previous assumption on $\mathcal{U}$), and the input to the trunk network also be bounded, as we restrict our analysis to a bounded domain.

Next, we define the semi-continuous version of $\mathcal{L}_{data} = \mathcal{L}_{data}(\eta, \theta, \{x_l\}_{l=1}^L)$ as follows:

$$\overline{\mathcal{L}}_{data}(\eta, \theta) = \int_{\mathcal{U}} \frac{|\Omega_m|}{L} \sum_{l=1}^L \left[ u_{\eta,\theta}(x_l) - u(x_l) \right] d\mu(\mathcal{U})$$

$$= \int_{\mathcal{U}} \sum_{i=1}^N \frac{1}{N} \frac{|\Omega_m|}{L} \sum_{l=1}^L \left[ u_{\eta,\theta}^{(i)}(x_l) - u^{(i)}(x_l) \right]^2 dx d(\mu(\mathcal{U}))^N,$$

where $u_{\eta,\theta}^{(i)}$ denotes the PI-DIONs solution for the $i$-th component.

We first quantify the difference between $\overline{\mathcal{L}}_{data}$ and $\mathcal{L}_{data}$. Note that $\overline{\mathcal{L}}_{data}(\eta, \theta)$ is bounded by $4R^2|\Omega_m|$, where $|\Omega_m|$ denotes the area of $\Omega_m$. Using this boundedness, along with Hoeffding's inequality, we derive the following lemma.

**Lemma 1.** *For any $\alpha > 0$ and $L \geq 64R^4|\Omega_m|^2 \log 2/\alpha^2$, the following inequality holds.*

$$\mathbb{P}\left( \{x_l\}_{l=1}^L \in \Omega_m, \{u^{(i)}\}_{i=1}^N \in \mathcal{U} \middle| \sup_{\eta,\theta} |\overline{\mathcal{L}}_{data}(\eta, \theta) - \mathcal{L}_{data}(\eta, \theta, \{x_l\}_{l=1}^L)| > \alpha \right)$$

$$\leq 2\mathbb{P}\left( \{x_l\}_{l=1}^{2L} \in \Omega_m, \{u^{(i)}\}_{i=1}^N \in \mathcal{U} \middle| \sup_{\eta,\theta} |\mathcal{L}_{data}(\eta, \theta, \{x_l\}_{l=L+1}^{2L}) - \mathcal{L}_{data}(\eta, \theta, \{x_l\}_{l=1}^L)| > \frac{\alpha}{2} \right)$$

*Proof.* We first focus on determining a lower bound for the right-hand side of the inequality. Since the event on the left-hand side includes the event on the right-hand side, we can write.

$$\mathbb{P}\left( \{x_l\}_{l=1}^{2L} \in \Omega_m, \{u^{(i)}\}_{i=1}^N \in \mathcal{U} \middle| |\mathcal{L}_{data}(\eta, \theta, \{x_l\}_{l=L+1}^{2L}) - \mathcal{L}_{data}(\eta, \theta, \{x_l\}_{l=1}^L)| > \frac{\alpha}{2} \right)$$

$$\geq \mathbb{P}\left( \{x_l\}_{l=L+1}^{2L} \in \Omega_m, \{u^{(i)}\}_{i=1}^N \in \mathcal{U} \middle| |\overline{\mathcal{L}}_{data}(\eta, \theta) - \mathcal{L}_{data}(\eta, \theta, \{x_l\}_{l=1}^L)| > \alpha \right) \cdot$$

$$\mathbb{P}\left( \{x_l\}_{l=1}^L \in \Omega_m, \{u^{(i)}\}_{i=1}^N \in \mathcal{U} \middle| |\overline{\mathcal{L}}_{data}(\eta, \theta) - \mathcal{L}_{data}(\eta, \theta, \{x_l\}_{l=1}^L)| < \frac{\alpha}{2} \right)$$

Now, we focus on bounding the last term on the right-hand side using Hoeffding's inequality, leveraging the boundedness of $\mathcal{L}_{data}$. Specifically, we have:

$$\mathbb{P}\left( \{x_l\}_{l=1}^L \in \Omega_m, \{u^{(i)}\}_{i=1}^N \in \mathcal{U} \middle| |\overline{\mathcal{L}}_{data}(\eta, \theta) - \mathcal{L}_{data}(\eta, \theta, \{x_l\}_{l=1}^L)| < \frac{\alpha}{2} \right)$$

$$\geq 1 - 2\exp\left( -\frac{L\alpha^2}{32R^4|\Omega_m|^2} \right)$$

This follows from the fact that each term on the left-hand side can be computed as:

$$|\overline{\mathcal{L}}_{data}(\eta, \theta) - \mathcal{L}_{data}(\eta, \theta, \{x_l\}_{l=1}^L)|$$

$$= \int_{\mathcal{U}} \sum_{i=1}^N \frac{1}{N} \left( \frac{|\Omega_m|}{L} \sum_{l=1}^L \left[ u_{\eta,\theta}^{(i)}(x_l) - u^{(i)}(x_l) \right]^2 - \int_{\Omega_m} \left[ u_{\eta,\theta}^{(i)}(x) - u^{(i)}(x) \right]^2 dx \right) d\mu(\mathcal{U})^N,$$

and

$$0 \leq \frac{|\Omega_m|}{L} \left[u_{\eta,\theta}^{(i)}(x_l) - u^{(i)}(x_l)\right]^2 \leq \frac{4R^2}{L}|\Omega_m|.$$

If $L \geq (64\log 2)R^4\Omega_m^2/\alpha^2$ holds, we obtain the desired inequality. $\qquad\square$

We denote the function space that can be represented by $u_{\eta,\theta}(x)$ as $\widetilde{\mathcal{U}}$. To quantify the complexity of this space, we define the covering number, which represents the number of elements in a set such that any arbitrary element can find a sufficiently close representative in that set. This covering number will be instrumental in deriving an upper bound for $|\widetilde{\mathcal{L}}_{data} - \mathcal{L}_{data}|$.

**Definition 1.** *For a given $\varepsilon > 0$, we define $\mathcal{N}(\varepsilon, \widetilde{\mathcal{U}})$ as follows. Let $S_c = \{(u_{\eta_1,\theta_1}), \cdots, (u_{\eta_S,\theta_S})\}$ be a set such that for any $u_{\eta,\theta} \in \widetilde{\mathcal{U}}$, there exists an element $u_{\eta_k,\theta_k}$ in set $S_c$ such that the following inequality holds.*

$$\sum_{i=1}^N \frac{1}{N} \sum_{l=1}^{2L} \frac{|\Omega_m|}{2L} \left| \left[u_{\eta_k,\theta_k}^{(i)}(x_l) - u^{(i)}(x_l)\right]^2 - \left[u_{\eta,\theta}^{(i)}(x_l) - u^{(i)}(x_l)\right]^2 \right| \leq \varepsilon$$

*$\mathcal{N}(\varepsilon, \tilde{U})$ is set to the minimum number of such $S$. Furthermore, we define $N_c(\varepsilon, \tilde{U})$ to be the supremum of $\mathcal{N}(\varepsilon, \tilde{U})$ over any choice of $\{u^{(i)}\}_{i=1}^N$ and $\{x_l\}_{l=1}^L$. Note that $N_c$ may be infinite.*

Next we introduce the subset $\Pi$ of the permutation group $S_{2L}$. An element $\sigma \in \Pi$ is defined such that for all $1 \leq l \leq L$, either $\sigma(l) = l$ and $\sigma(l + L) = l + L$, or $\sigma(l) = l + L$ and $\sigma(l + L) = l$. With this definition, the following lemma holds.

**Lemma 2.** *Suppose that $N_c$ is the covering number for $\alpha/8$ where $S_c = \{(u_{\eta_1,\theta_1}), \cdots, (u_{\eta_{N_c},\theta_{N_c}})\}$ is a covering set for $\tilde{\mathcal{U}}$. Then the following inequality holds:*

$$\mathbb{P}\left( \{x_l\}_{l=1}^{2L} \in \Omega_m, \{u^{(i)}\}_{i=1}^N \in \mathcal{U} \,\middle|\, \sup_{\eta,\theta} |\mathcal{L}_{data}(\eta, \theta, \{x_l\}_{l=L+1}^{2L}) - \mathcal{L}_{data}(\eta, \theta, \{x_l\}_{l=1}^L)| > \frac{\alpha}{2} \right)$$

$$= \mathbb{P}\left( \sigma \in \Pi \,\middle|\, \sup_{\eta,\theta} |\mathcal{L}_{data}(\eta, \theta, \{x_{\sigma(l)}\}_{l=L+1}^{2L}) - \mathcal{L}_{data}(\eta, \theta, \{x_{\sigma(l)}\}_{l=1}^L)| > \frac{\alpha}{2} \right)$$

$$\leq \sum_{k=1}^{N_c} \mathbb{P}\left( \sigma \in \Pi \,\middle|\, |\mathcal{L}_{data}(\eta_k, \theta_k, \{x_{\sigma(l)}\}_{l=L+1}^{2L}) - \mathcal{L}_{data}(\eta_k, \theta_k, \{x_{\sigma(l)}\}_{l=1}^L)| > \frac{\alpha}{4} \right)$$

*Proof.* For a permutation $\sigma$ and parameters $\eta, \theta$, assume the following inequality holds.

$$|\mathcal{L}_{data}(\eta, \theta, \{x_{\sigma(l)}\}_{l=L+1}^{2L}) - \mathcal{L}_{data}(\eta, \theta, \{u_{\sigma(l)}\}_{l=1}^L)| > \frac{\alpha}{2}$$

We claim that there exists at least one element in $S_c$ such that:

$$|\mathcal{L}_{data}(\eta_k, \theta_k, \{x_{\sigma(l)}\}_{l=L+1}^{2L}) - \mathcal{L}_{data}(\eta_k, \theta_k, \{x_{\sigma(l)}\}_{l=1}^L)| > \frac{\alpha}{4}$$

Suppose, on the contrary, that no such element exists. By the definition of covering number, we can find a $\theta_k, \eta_k$ such that:

$$\frac{\Omega_m}{2L} \sum_{l=1}^{2L} \frac{1}{N} \sum_{i=1}^N \left| \left[u_{\eta_k,\theta_k}^{(i)}(x_l) - u_i(x_l)\right]^2 - \left[u_{\eta,\theta}^{(i)}(x_l) - u_i(x_l)\right]^2 \right| \leq \frac{\alpha}{4}.$$

Now, observe that the following inequality holds, where the last inequality is obtained by applying the standard triangle inequality.

$$\frac{\Omega_m}{2L}\sum_{l=1}^{2L}\frac{1}{N}\sum_{i=1}^{N}\left|\left[u_{\eta_k,\theta_k}(x_l)-u(x_l)\right]^2-\left[u_{\eta,\theta}(x_l)-u(x_l)\right]^2\right|$$

$$=\frac{\Omega_m}{2L}\sum_{l=1}^{L}\frac{1}{N}\sum_{i=1}^{N}\left|\left[u_{\eta_k,\theta_k}^{(i)}(x_{\sigma(l)})-u^{(i)}(x_{\sigma(l)})\right]^2-\left[u_{\eta,\theta}^{(i)}(x_{\sigma(l)})-u^{(i)}(x_{\sigma(l)})\right]^2\right|$$

$$+\frac{\Omega_m}{2L}\sum_{l=L+1}^{2L}\frac{1}{N}\sum_{i=1}^{N}\left|\left[u_{\eta_k,\theta_k}^{(i)}(x_{\sigma(l)})-u^{(i)}(x_{\sigma(l)})\right]^2-\left[u_{\eta,\theta}^{(i)}(x_{\sigma(l)})-u^{(i)}(x_{\sigma(l)})\right]^2\right|$$

$$=\frac{1}{2}|\mathcal{L}_{data}(\eta,\theta,\{x_l\}_{l=L+1}^{2L})-\mathcal{L}_{data}(\eta,\theta,\{x_{\sigma(l)}\}_{l=1}^{L})|$$

$$-\frac{1}{2}|\mathcal{L}_{data}(\eta_k,\theta_k,\{x_{\sigma(l)}\}_{l=L+1}^{2L})-\mathcal{L}_{data}(\eta_k,\theta_k,\{x_{\sigma(l)}\}_{l=1}^{L})|.$$

Since the first term is bounded by $\alpha/8$ and the last term is strictly greater than $\alpha/4-\alpha/8=\alpha/8$, we arrive at a contradiction. Therefore, the claim holds. $\square$

Finally, we derive the upper bound for the result in the previous lemma using hoeffding's inequality with zero mean.

**Lemma 3.** *For any $\eta,\theta$, the following inequality holds.*

$$\mathbb{P}\left(\sigma\in\Pi\middle|\,|\mathcal{L}_{data}(\eta,\theta,\{x_{\sigma(l)}\}_{l=L+1}^{2L})-\mathcal{L}_{data}(\eta,\theta,\{x_{\sigma(l)}\}_{l=1}^{L})|>\frac{\alpha}{4}\right)\leq 2\exp\left(-\frac{\alpha^2 NL}{512|\Omega_m|^2 R^4}\right)$$

*Proof.* By definition of $\Pi$, we observe that the condition:

$$|\mathcal{L}_{data}(\eta,\theta,\{x_{\sigma(l)}\}_{l=L+1}^{2L})-\mathcal{L}_{data}(\eta,\theta,\{x_{\sigma(l)}\}_{l=1}^{L})|>\frac{\alpha}{4}$$

can be written as:

$$\mathbb{P}\left(\sigma\in\Pi\middle|\,|\frac{\Omega_m}{L}\sum_{l=1}^{L}\frac{1}{N}\sum_{i=1}^{N}\left[u_{\eta,\theta}^{(i)}(x_{\sigma(l+L)})-u_i(x_{\sigma(l+L)})\right]^2-\left[u_{\eta,\theta}^{(i)}(x_{\sigma(l)})-u^{(i)}(x_{\sigma(l)})\right]^2|\right)>\frac{\alpha}{4}\right)$$

This can be further transformed as:

$$=\mathbb{P}\left(\sigma_i\in\left\{-\frac{1}{2},\frac{1}{2}\right\}\text{ for each }i\middle|\frac{\Omega_m}{LN}|\sum_{l=1}^{L}\sum_{i=1}^{N}\sigma_i\left(\left[u_{\eta,\theta,i}(x_{l+L})-u_i(x_{l+L})\right]^2-\left[u_{\eta,\theta,i}(x_l)-u_i(x_l)\right]^2\right)|d\mu(\mathcal{U})>\frac{\alpha}{4}\right)$$

$$\leq 2\exp\left(-NL\frac{\alpha^2}{512|\Omega_m|^2 R^4}\right)$$

where the last inequality is obtained by Hoeffding's inequality. This holds because, by the boundedness assumption on $u_{\eta,\theta}$, the following bound applies:

$$-4R^2\leq\sigma^{(i)}(\left[u_{\eta,\theta}^{(i)}(x_{l+L})-u^{(i)}(x_{l+L})\right]^2-\left[u_{\eta,\theta}^{(i)}(x_l)-u^{(i)}(x_l)\right]^2)|\leq 4R^2$$

Thus, the result follows. $\square$

**Proposition 2.** *Now if $L$ is larger than $64R^4|\Omega_m|^2\log 2/\alpha^2$, then the following inequality holds.*

$$\mathbb{P}\left(\{x_l\}_{l=1}^{L}\in\Omega_m,\{u^{(i)}\}_{i=1}^{N}\in\mathcal{U}^N\middle|\sup_{\eta,\theta}|\overline{\mathcal{L}}_{data}(\eta,\theta)-\mathcal{L}_{data}(\eta,\theta,\{x_l\}_{l=1}^{L})|>\alpha\right)$$

$$\leq 4N_c exp\left(-\frac{\alpha^2 NL}{512|\Omega_m|^2 R^4}\right)$$

For the proof, we seek to find a uniform bound for the following term.

$$\sup_{\eta,\theta} \int_{\mathcal{U}} |\frac{\Omega_m}{L} \sum_{l=1}^{L} \big[u_{\eta,\theta}(x_l) - u(x_l)\big]^2 - \int_{\Omega_m} \big[u_{\eta,\theta}(x) - u(x)\big]^2 dx| d\mu(\mathcal{U}).$$

We proceed similarly to the previous analysis and first establish the following lemma. We omit the proof here, as it can be shown similarly to previous ones.

**Lemma 4.** *For fixed parameters $\eta$ and $\theta$, the following inequality holds if $L \geq \frac{64R^4\Omega_m^2 \log 2}{\alpha^2}$.*

$$\mathbb{P}\left(\sup_{\eta,\theta} \int_{\mathcal{U}} |\frac{\Omega_m}{L} \sum_{l=1}^{L} \big[u_{\eta,\theta}(x_l) - u(x_l)\big]^2 - \int_{\Omega_m} \big[u_{\eta,\theta}(x) - u(x)\big]^2 dx| d\mu > \alpha\right)$$

$$\leq 2\mathbb{P}\left(\sup_{\eta,\theta} \int_{\mathcal{U}} |\frac{\Omega_m}{L} \sum_{l=1}^{L} \big[u_{\eta,\theta}(x_l) - u(x_l)\big]^2 - \frac{\Omega_m}{L} \sum_{l=1}^{L} \big[u_{\eta,\theta}(x_{l+L}) - u(x_{l+L})\big]^2| d\mu > \frac{\alpha}{2}\right),$$

*where $\{x_{l+L}\}_{l=1}^{L}$ are sampled from $\Omega$ using the uniform distribution.*

Suppose that $\Pi$ is defined as in the previous. We can now derive the following two lemmas using the same calculations as before.

**Lemma 5.** *Suppose that $N_c$ is covering number for $\frac{\alpha}{8}$. Then, the following inequality holds.*

$$\mathbb{P}(\sigma \in \Pi \bigg| \sup_{\eta,\theta} \int_{\mathcal{U}} |\frac{\Omega_m}{L} \sum_{l=1}^{L} \big[u_{\eta,\theta}(x_{\sigma(l)}) - u(x_{\sigma(l)})\big]^2 - \frac{\Omega_m}{L} \sum_{l=1}^{L} \big[u_{\eta,\theta}(x_{\sigma(l+L)}) - u(x_{\sigma(l+L)})\big]^2| d\mu > \frac{\alpha}{2})$$

$$\leq \sum_{k=1}^{N_c} \mathbb{P}\left(\int_{\mathcal{U}} |\frac{\Omega_m}{L} \sum_{l=1}^{L} \big[u_{\eta_k,\theta_k}(x_{\sigma(l)}) - u(x_{\sigma(l)})\big]^2 - \frac{\Omega_m}{L} \sum_{l=1}^{L} \big[u_{\eta_k,\theta_k}(x_{\sigma(l+L)}) - u(x_{\sigma(l+L)})\big]^2| d\mu > \frac{\alpha}{4}\right)$$

**Lemma 6.** *For any $\sigma \in \pi$, the following inequality holds.*

$$\sum_{k=1}^{N_c} \mathbb{P}\left(\int_{\mathcal{U}} |\frac{\Omega_m}{L} \sum_{l=1}^{L} \big[u_{\eta_k,\theta_k}(x_{\sigma(l)}) - u(x_{\sigma(l)})\big]^2 - \frac{\Omega_m}{L} \sum_{l=1}^{L} \big[u_{\eta_k,\theta_k}(x_{\sigma(l+L)}) - u(x_{\sigma(l+L)})\big]^2| d\mu > \frac{\alpha}{4}\right)$$

$$\leq 2\exp\left(-\frac{\alpha^2 NL}{512R^4|\Omega_m|^2}\right)$$

Finally, we can derive the following proposition, which shows that $\widetilde{\mathcal{L}}_{data}$ is close to $\mathcal{L}_{data}$ with high probability.

**Proposition 3.** *If $L$ is larger than $64R^4|\Omega_m|^2 \log 2/\alpha^2$, then the following inequality holds.*

$$\mathbb{P}\left(\{x_l\}_{l=1}^{L} \in \Omega_m, \{u^{(i)}\}_{i=1}^{N} \in \mathcal{U}^N \bigg| \sup_{\eta,\theta} |\widetilde{\mathcal{L}}_{data}(\eta,\theta) - \mathcal{L}_{data}(\eta,\theta,\{x_l\}_{l=1}^{L})| > 2\alpha\right)$$

$$\leq 8N_c exp\left(-\frac{\alpha^2 NL}{512|\Omega_m|^2 R^4}\right)$$

*Proof.* By triangle inequality, we have the following.

$$\mathbb{P}\left(\{x_l\}_{l=1}^{L} \in \Omega_m, \{u^{(i)}\}_{i=1}^{N} \in \mathcal{U}^N \bigg| \sup_{\eta,\theta} |\widetilde{\mathcal{L}}_{data}(\eta,\theta) - \mathcal{L}_{data}(\eta,\theta,\{x_l\}_{l=1}^{L})| > 2\alpha\right)$$

$$\leq \mathbb{P}\left(\{x_l\}_{l=1}^{L} \in \Omega_m, \{u^{(i)}\}_{i=1}^{N} \in \mathcal{U}^N \bigg| \sup_{\eta,\theta} |\widetilde{\mathcal{L}}_{data}(\eta,\theta) - \mathcal{L}_{data}(\eta,\theta,\{x_l\}_{l=1}^{L})| > \alpha\right)$$

$$+ \mathbb{P}\left(\{x_l\}_{l=1}^{L} \in \Omega_m, \{u^{(i)}\}_{i=1}^{N} \in \mathcal{U}^N \bigg| \sup_{\eta,\theta} |\overline{\mathcal{L}}_{data}(\eta,\theta) - \mathcal{L}_{data}(\eta,\theta,\{x_l\}_{l=1}^{L})| > \alpha\right)$$

By applying Lemma 4, 5, and 6, along with Proposition 3, we can derive the desired result. $\square$

## D.2   PROOF OF THEOREM 2

Although $\widetilde{\mathcal{L}}_{physics}$ minimizes both two terms for the interior and boundary, the proof of Theorem 2 follows similarly. Specifically, we have:

$$\tilde{\mathcal{L}}_{physics} = \int_{\mathcal{U}}\int_{\Omega}\left[\mathcal{N}(u_{\eta,\theta}(x), s_{\zeta,\theta}(x))\right]^2 dxd\mu(\mathcal{U}) + \int_{\mathcal{U}}\int_{\partial\Omega}\left[\mathcal{B}(u(x))\right]^2 dxd\mu(\mathcal{U}).$$

The only additional consideration is verifying boundedness of the derivative $\mathcal{N}(u_{\eta,\theta}(x), s_{\zeta,\theta}(x))$ and the definition of the covering number. Once we redefine the covering number and assume that $\mathcal{N}(u,s)$ and $\mathcal{N}(u_{\eta,\theta}, s_{\zeta,\theta})$ are bounded by $R_{\mathcal{N}}$, and $\mathcal{B}(u,s)$ and $\mathcal{B}(u_{\eta,\theta})$ are bounded by $R_{\mathcal{B}}$, we can derive the Theorem 2. Let $\widetilde{\mathcal{S}}$ denote the set of functions comprising $s_{\zeta,\theta}$.

**Definition 2.** *For a given $\varepsilon > 0$, we define $\mathcal{N}(\varepsilon, \widetilde{\mathcal{U}} \times \widetilde{\mathcal{S}})$ as follows. Suppose we find a set $A = \{(u_{\eta_1,\theta_1}, s_{\eta_1,\zeta_1}), \cdots, (u_{\eta_a,\theta_a}, s_{\eta_a,\zeta_a})\}$ such that for any $u_{\eta,\theta} \in \widetilde{\mathcal{U}}$ and $s_{\zeta,\theta} \in \widetilde{\mathcal{S}}$, there exists an element $u_{\eta_k,\theta_k}, s_{\eta_k,\theta_k}$ in set $A$ such that the following inequality holds:*

$$\sum_{i=1}^{N}\frac{1}{N}\sum_{l=1}^{2K}\frac{|\Omega|}{2K}\left|\left[\mathcal{N}(u^{(i)}_{\eta_k,\theta_k}(x_l))\right]^2 - \left[\mathcal{N}(u^{(i)}_{\eta,\theta}(x_l))\right]^2\right| \leq \varepsilon, \quad x_l \in \Omega.$$

*$\mathcal{N}(\varepsilon, \tilde{U})$ is defined as the minimum number of such $S$. The covering number $N_{\mathcal{N}}$ is defined as the supremum of $S$ over any given $\{u^{(i)}\}_{i=1}^N$ and $\{x_l\}_{l=1}^L$. Similarly, $N_{\mathcal{B}}$ is defined using the inequality:*

$$\sum_{i=1}^{N}\frac{1}{N}\sum_{l=1}^{2M}\frac{|\partial\Omega|}{2M}\left|\left[\mathcal{B}(u^{(i)}_{\eta_k,\theta_k}(x_l))\right]^2 - \left[\mathcal{B}(u^{(i)}_{\eta,\theta}(x_l))\right]^2\right| \leq \varepsilon, \quad x_l \in \partial\Omega.$$

With the above covering number, we can derive a bound on the difference between $\mathcal{L}_{physics}$ and $\widetilde{\mathcal{L}}_{physics}$. Since the computation follows a similar procedure, we omit the detailed proof.

**Proposition 4.** *Now if $K$ and $M$ are larger than $4R^4|\Omega|^2\log 2/\alpha^2$ and $4R^4|\partial\Omega|^2\log 2/\alpha^2$, then the following inequality holds.*

$$\mathbb{P}\left(\{x_k\}_{k=1}^K \in \Omega, \{x_j\}_{j=1}^M \in \partial\Omega, \{u^{(i)}\}_{i=1}^N \in \mathcal{U}\Big| \sup_{\eta,\theta}|\widetilde{\mathcal{L}}_{physics}(\eta,\theta) - \mathcal{L}_{physics}(\eta,\theta,\{x_k\}_{k=1}^K,\{x_j\}_{j=1}^M)| > 4\alpha\right)$$

$$\leq 8N_c exp\left(-\frac{\alpha^2 NK}{32|\Omega|^2 R_{\mathcal{N}}^4}\right) + 8N_c exp\left(-\frac{\alpha^2 NM}{32|\partial\Omega|^2 R_{\mathcal{B}}^4}\right).$$

The key part of the proof involves bounding the lefthand side of the inequality in the proposition. For both terms $\int_{\mathcal{U}}\int_{\Omega}\left[\mathcal{N}(u_{\eta,\theta}(x), s_{\zeta,\theta}(x))\right]^2 dxd\mu(\mathcal{U})$ and $\int_{\mathcal{U}}\int_{\partial\Omega}\left[\mathcal{B}(u(x))\right]^2 dxd\mu(\mathcal{U})$, the previous approach in the proof of Theorem 3 can be applied to each part.

$$\mathbb{P}\left(\{x_k\}_{k=1}^K \in \Omega, \{x_j\}_{j=1}^M \in \partial\Omega, \{u^{(i)}\}_{i=1}^N \in \mathcal{U}\Big| \sup_{\eta,\theta}|\widetilde{\mathcal{L}}_{physics}(\eta,\theta) - \mathcal{L}_{physics}(\eta,\theta,\{x_k\}_{k=1}^K,\{x_j\}_{j=1}^M)| > 4\alpha\right)$$

$$\leq \mathbb{P}\left(\{x_k\}_{k=1}^K \in \Omega, \{u^{(i)}\}_{i=1}^N \in \mathcal{U}\Big| \sup_{\eta,\theta}|\widetilde{\mathcal{L}}_{physics}(\eta,\theta) - \mathcal{L}_{physics}(\eta,\theta,\{x_k\}_{k=1}^K,\{x_j\}_{j=1}^M)| > 2\alpha\right)$$

$$+ \mathbb{P}\left(\{x_j\}_{j=1}^M \in \partial\Omega, \{u^{(i)}\}_{i=1}^N \in \mathcal{U}\Big| \sup_{\eta,\theta}|\widetilde{\mathcal{L}}_{physics}(\eta,\theta) - \mathcal{L}_{physics}(\eta,\theta,\{x_k\}_{k=1}^K,\{x_j\}_{j=1}^M)| > 2\alpha\right),$$

## D.3   PROOF OF THEOREM 3

We now provide the proof for Theorem 3, which guarantees low prediction error for arbitrary functions $u$ and $s$ in the function space.

By applying Markov's inequality, we can ensure that samples $(u, s)$ from the measure $(\mu(u), \mu(s))$ approximately satisfy the governing equation, boundary condition and partial measurements consis-

tency. Specifically, any for given $\alpha > 0$, the following probability bound holds:

$$\mathbb{P}(u \in \mathcal{U} | \int_{\Omega_m} \left[ u_{\eta,\theta}(x) - u(x) \right]^2 dx + \int_{\Omega} \left[ \mathcal{N}(u_{\eta,\theta}(x), s_{\zeta,\theta}(x)) \right]^2 dx + \int_{\partial\Omega} \left[ \mathcal{B}(u(x)) \right]^2 dx \leq \alpha)$$
$$\geq 1 - (\tilde{\mathcal{L}}_{physics} + \tilde{\mathcal{L}}_{data})/\alpha.$$

Furthermore, with probability at least $1 - 2\delta$, we can bound the combined empirical losses $\tilde{\mathcal{L}}_{physics} + \tilde{\mathcal{L}}_{data}$ by the true losses $\mathcal{L}_{physics} + \mathcal{L}_{data}$ as follows:

$$\widetilde{\mathcal{L}}_{physics} + \widetilde{\mathcal{L}}_{data} \leq \mathcal{L}_{physics} + \mathcal{L}_{data} + 2\epsilon.$$

Recalling stability estimates for a single element in $\mathcal{U}$, we can conclude that, with probability at least $1 - (\tilde{\mathcal{L}}_{physics} + \tilde{\mathcal{L}}_{data})/\alpha$, the following error bound holds for $u \in \mathcal{U}$.

$$\|u_{\eta,\theta} - u\|_{L^2(\Omega)} + \|s_{\zeta,\theta} - s\|_{L^2(\Omega)} \leq \alpha.$$

Finally, by choosing $\alpha = \sqrt{\epsilon}$, we obtain the desired probabilistic bounds on the solution error.

### D.4    PROOF OF PROPOSITION 1

We first refer to a relevant theorem which states DeepONet is a universal approximator, allowing for close approximation with a sufficient number of parameters.

**Theorem 4.** *(Chen & Chen (1995)) Consider the case where $\mathcal{X}$ is Banach space and $K$ is a compact subset of $\mathcal{X}$. Suppose that an operator $\mathcal{G} : \mathcal{U} \to \mathcal{S}$ is continuous where $\mathcal{U}$ is a compact subset of the infinite-dimensional function space $C(\mathcal{K}; \mathbb{R})$ and $\mathcal{S}$ consists of the function whose domain is a compact subset $L$ of $\mathbb{R}^d$, Then for any $\epsilon > 0$, there exists the unstacked DeepONet with the shallow branch net $\beta$ and trunk net $\tau$ such that*

$$|\mathcal{G}(u)(y) - \langle \beta(u(x_1), \cdots, u(x_m); \theta_\beta), \tau(y; \theta_\tau) \rangle| < \epsilon,$$

*for all $u \in \mathcal{U}$ and $y \in \mathcal{Y}$.*

While one might argue that the above theorem only applies to compact subsets of function spaces, making it insufficient for learning operators between infinite-dimensional spaces, it is adequate for minimizing the discretized loss $\mathcal{L} = \mathcal{L}_{physics} + \mathcal{L}_{data}$, which is computed using finite samples and grid points. Moreover, DeepONet is a universal approximator for continuous operator. In our case, we address an inverse problem that often satisfies a stability estimate, ensuring continuity for both the solution operator and inverse operator. The detailed result is as follows.

**Proposition 5.** *(Universal Approximation Theorem for Operator). Assume that the branch network and trunk network in PI-DIONs have continuous, non-polynomial activation functions. Suppose that our inverse problem has the following stability estimate.*

$$\|u_2 - u_1\|_{L^2(\Omega)} + \|s_2 - s_1\|_{L^2(\Omega)} \leq \|u_2|_{\Omega_m} - u_1|_{\Omega_m}\|_{L^2(\Omega_m)}.$$

*For any given input-output dataset $\{(u^{(i)}|_{\Omega_m}, s^{(i)})\}_{i=1}^N$ and for any $\varepsilon > 0$ there exist appropriate parameters $\eta, \zeta$, and $\theta$ for PI-DIONs such that*

$$\mathcal{L}_{data} < \epsilon.$$

*Proof.* Consider the dataset $\{(u^{(i)}|_{\Omega_m}, u^{(i)}, s^{(i)})\}_{i=1}^N$ used for training. We aim to approximate the operator $\mathcal{G} : u|_{\Omega_m} \to (u, s)$ using PI-DION. Note that $\mathcal{G}|_{\{u^{(i)}|\partial\Omega_m\}_{i=1}^N}$ is a continuous operator between $L^2(\Omega)$ and $\{L^2(\Omega)\}^2$. Furthermore, $\{u^{(i)}|\partial\Omega_m\}_{i=1}^N$ is a compact set, as it is finite. For any $1 > \epsilon > 0$, by Theorem 4, there exists PI-DIONs with $\eta, \zeta, \theta$ such that:

$$|\mathcal{G}(u^{(i)}|_{\Omega_m})(x) - (u_{\eta,\theta}, s_{\zeta,\theta})|^2 \leq |\mathcal{G}(u^{(i)}|_{\Omega_m})(x) - (u_{\eta,\theta}, s_{\zeta,\theta})| < \epsilon, \quad \forall x \in \Omega, \forall i \in \{1, 2, \cdots, N\}.$$

This implies that $\mathcal{L}_{data} < \epsilon$. $\qquad \qquad \square$

The previous proof demonstrates that we can guarantee the reduction of $\mathcal{L}_{data}$, and therefore $\widetilde{\mathcal{L}}_{data}$. However, providing the reduction of $\mathcal{L}_{physics}$ requires a slightly different approach, relying on the universal approximation theorem for neural networks. The details are as follows.

**Proof of proposition 1**

Consider the dataset $\{(u^{(i)}|_{\Omega_m}, u^{(i)}, s^{(i)})\}_{i=1}^N$ used for training. We define three different neural networks $\beta_\eta, \beta_\zeta, \tau_\theta$ where $\beta_\eta(\{u^{(i)}(x_l)\}_{l=1}^L)$ and $\beta_\zeta(\{u^{(i)}(x_l)\}_{l=1}^L) \sim (0, \cdots, 1, \cdots, 0, 0 \cdots, 0)$ and $(0, \cdots, 0, 0, \cdots, 1, \cdots, 0) \in \mathbb{R}^{2N}$ with the vectors in $\mathbb{R}^{2N}$ having a 1 in $(i)$-th and $(i+N)$-th coordinate respectively. Meanwhile, the network $\tau$ approximates $(u_1(x), \cdots, u_N(x), s_(x), \cdots, s_N(x))$.

By the universal approximation theorem for neural networks, for any multi-index $\alpha$, we have:

$$\sum_{|\alpha|<=n} \frac{\partial}{\partial^\alpha x} |\tau(x) - (u_1(x), s_1(x), \cdots, u_N(x), c_N(x))| \leq \epsilon.$$

If differential operator $\mathcal{N}$ is linear, it is straightforward to show that:

$$\mathcal{L}_{physics} + \mathcal{L}_{data} \leq \varepsilon$$

If $\mathcal{N}$ is nonlinear, involving product terms, the desired result can still be derived by applying the triangle inequality, assuming the boundedness of $u$. Thus, for both linear and nonlinear cases, the loss $\mathcal{L}_{physics} + \mathcal{L}_{data}$ can be made arbitrarily small, completing the proof.

