# OpenReview forum: "Physics-Informed Deep Inverse Operator Networks for Solving PDE Inverse Problems"
_ICLR.cc/2025/Conference — ICLR 2025 Poster_

### Official Review · Reviewer_knEo · 2024-10-23

**Soundness:** 2
**Presentation:** 3
**Contribution:** 2
**Rating:** 6
**Confidence:** 3

**Summary:**

This paper introduces Physics-Informed Deep Inverse Operator Networks (PI-DIONs) for solving PDE-based inverse problems without the need for labeled data. The paper extends existing stability estimates from inverse problem literature to the operator learning framework, ensuring the robustness and generalizability of PI-DIONs across the entire function space and domain.

**Strengths:**

1. This paper provides a solid theoretical foundation for the proposed PI-DIONs.
2. The proposed method demonstrates practicality and efficiency in addressing PDE-based inverse problems without the need for labeled data.

**Weaknesses:**

1. The contribution lacks novelty. The architecture relies on relatively simple components, such as CNNs and MLPs for the branch and trunk networks. It doesn't introduce significant advancements beyond well-established methods.
2. The baselines used for comparison, such as DeepONet and FNO, are somewhat dated. The paper would benefit from comparisons with more recent and state-of-the-art methods to better demonstrate the model's competitiveness.
3. The experimental evaluation is limited in range. Conducting experiments on a broader range of benchmarks would strengthen the validation of the proposed method's effectiveness across diverse problems.

**Questions:**

See above.

---

> ### Author Response · Authors · 2024-11-15
> **Response to reviewer knEo**
>
> Thank you for clearly summarizing our contributions and highlighting the strengths of our work.
>
> Response to Weaknesses:
>
> 1. To the best of our knowledge, PI-DIONs and PI-DIONs-v0 (in Appendix A) are the first physics-informed operator networks specifically designed for inverse problems. Due to structural limitations, existing architectures such as DeepONet and FNO cannot incorporate physics loss for inverse problems. A detailed description of the limitations are presented in Chapter 2 (lines 105-108). The relatively simple network utilized in this work was chosen to highlight the novelty of the architecture and loss function while ensuring a fair comparison, which we encourage you to consider.
> In addition to the architectural novelty, we provide rigorous theoretical justification for our method, as detailed in Section 3. Specifically, we extend stability estimates to the operator learning framework, demonstrating that sufficiently small $\mathcal{L}$ implies a small prediction error for both the solution and target functions. Furthermore, we present a universal approximation theorem for PI-DIONs, which guarantees that the loss function can be reduced to an arbitrarily small value. These theoretical contributions enhance the significance of our work, and to the best of our knowledge, these are the first theoretical results for inverse operator learning.
>
> 2. Thank you for your valuable feedback regarding the comparison with the baseline. Solving forward and inverse problems using physics-informed methods, such as differential equations, is a relatively recent area of research. While NIO is a promising approach, a direct comparison with our model is challenging because it relies on fully supervised learning of the operator-to-function mapping. In contrast, our model can be trained without direct supervision on the target function, and to the best of our knowledge, there is no existing architecture that aligns with this approach. Nevertheless, we can also enhance our methodology by incorporating DeepONet’s advanced model into PI-DIONs
> Since NIO utilizes DeepONet as a critical baseline, we selected the same model for comparison. If you could suggest relevant references addressing inverse problems with physics-informed methods, we would be glad to conduct further comparative experiments.
>
> 3. The benchmark problems are widely used ones in physics-informed machine learning literature. In particular, even PINN approaches for solving the inverse source problem for the reaction-diffusion and the Helmholtz equations have been proposed very recently [2,3]. Our work extends recent works to the operator learning framework, where PI-DION rapidly predicts both the solution and target function from the partial measurements of the solution. Moreover, even in the absence of formal stability estimates, PI-DION consistently achieves accurate approximations for both solutions and target functions in the Darcy flow problem, suggesting its potential applicability to various differential equations. Furthermore, we have conducted additional experiments, including sensitivity analysis, detailed in Appendix C. Compared to PINNs, it offers significantly faster computation, while maintaining slightly larger but negligible relative errors.
>
>
> [1] Roberto Molinaro, Yunan Yang, Bj¨orn Engquist, and Siddhartha Mishra. Neural inverse operators for solving pde inverse problems. 2023.
>
> [2] Hui Zhang and Jijun Liu. Solving an inverse source problem by deep neural network method with convergence and error analysis. 2023. Inverse Problems
>
> [3] Mengmeng Zhang, Qianxiao Li, and Jijun Liu. On stability and regularization for data-driven solution of parabolic inverse source problems. 2023. Journal of Computational Physics

---

> > ### Comment · Reviewer_knEo · 2024-11-22
> >
> > I appreciate the authors' response and additional experiments. I have raised my score.

---

> > > ### Author Response · Authors · 2024-11-22
> > > **Thank you**
> > >
> > > We sincerely appreciate your thoughtful consideration. Please feel free to reach out if you have any further questions or would like to discuss anything further.

---

### Official Review · Reviewer_3Fuq · 2024-10-31

**Soundness:** 3
**Presentation:** 2
**Contribution:** 2
**Rating:** 5
**Confidence:** 5

**Summary:**

This paper proposes an architecture called Physics-Informed Deep Inverse Operator Networks (PI-DIONs), which can learn the solution operator of PDE-based inverse problems without labeled training data. The architecture  of PI-DIONs is based on DeepONet, and trained with both the physics-infomred loss and data reconstruction loss. The stability estimates established in the inverse problem literature are extended to the operator learning framework. Experiments are conducted to demonstrate the effectiveness of PI-DIONs in learning the solution operators of the inverse problems without the need for labeled data.

**Strengths:**

1. The integration of physics-informed losses into an inverse problem framework based on operator learning is novel, and in principle PI-DIONs can solve the inverse problems (at least in scenarios mentioned in experiments) fast and without the need for labeled data.
2. Theoretical analysis of the stability estimates is provided.

**Weaknesses:**

1. Line 243, "where the term ∥f − f^\star∥L2(Ωm) in the righthand side", there is no such term there. Please clarify the equation in line 242 and include all terms on the right-hand side of the equation.
2. It seems that the input to the reconstruction and inverse branch networks is fixed in shape, corresponding to the partial measurement with given geometry. The observed data in PINNs can have variable count and locations. Please discuss how PI-DIONs might be adapted to handle variable measurement geometries and if there are any limitations on the types of measurement setups it can handle.
3. In the experiments, PI-DIONs are compared with purely data-driven DeepONet and FNO, which both did not take physics information into account. If possible, please include comparisons with PINNs in the experiments, since both your PI-DIONs and PINNs are physics-informed methods for inverse problems.
4. The simultaneous training of physics-informed losses for 1000 samples is a difficult task (similar to train 1000 PINNs simultaneously). I am curious about the training difficulties encountered. Please provide specific details on training time, hardware used, and any convergence challenges encountered. If possible, please also include an ablation study on the effect of sample size on PI-DIONs' performance since smaller sample size may lead to easier optimization.
5. The theoretical analysis on stability estimate is extended from existing key results that considered the single element case.
6. Please provide a clear definition of u in line 152 and describe its relationship with partial measurement. In line 456, it is better to write "f(x,y) = 100x(1 − x)y(1 − y) ", so does line 450.

Considering the above weaknesses, I give a score of 3 to the current version of this paper.

**Questions:**

1. DeepONet and FNO are used for forward problems traditionally, how did they deal with inverse problems in your experiments?
2. How is the labeled training target f mentioned in line 399 used?  The loss for target f is absent in line 152.

---

> ### Author Response · Authors · 2024-11-15
> **Response to reviewer 3Fuq**
>
> Thank you for clearly summarizing our contributions and highlighting the strengths of our work.
>
> Response to Weaknesses:
>
> 1. Thank you for the helpful feedback regarding the term on the right-hand side in Line 242. To improve clarity, we have revised the explanation by bringing the term $\Vert f-f^*\Vert_{L^2(\Omega_m)}$ to the forefront of inequality. Additionally, we have enhanced the logical flow by adding more details to better connect the inequalities in Lines 233-237 with the final inequality in Lines 242-244. We appreciate your effort in identifying the potential source of confusion.
>
> 2. Thank you for your insightful comment regarding inputs with variable counts and locations. The limitation in handling such inputs arises from PI-DION's reliance on the fundamental DeepONet framework. However, recent advancements, such as Variable Input Deep Operator Network [1] or GraphDeepONet [2] address this issue through aggregation functions with softmax. Incorporating these methodologies into the branch network could effectively resolve the above limitation. We have included this promising direction in the discussion section. We sincerely appreciate your valuable feedback.
>
> 3. Thank you for suggesting a comparison with PINNs. As highlighted in Section 2, this study was motivated by the limitation that physics-informed loss functions cannot be directly applied to inverse problems in operator learning methods like DeepONet and FNO. To address this, we developed the novel architecture presented in Figure 1.
> While PINNs are effective for inverse problems, they require retraining from scratch for each new set of partial measurements, leading to substantial computational costs. In contrast, PI-DIONs enable real-time inference after a single training phase. As shown in Table 5 (Appendix C.1), PI-DIONs achieve significantly faster inference times than PINNs, with only a slight increase in relative error, which is negligible (see Table 4 in Appendix C.1).
>
> 4. Thank you for suggesting ablation studies to further justify our model. Training details, including hardware specifications, are provided in Appendix B. In response to your feedback, we have now included training times in Table 5 (Appendix C.1) and the prediction errors for different sample sizes of 100, 500, 1000, and 2000 in Table 7 (Appendix C.2). As observed, approximately 1000 samples are required to achieve a relative error of about 1% for the target function, although smaller sample sizes may simplify optimization.
>
> 5. Thank you for clarifying the theoretical contribution of our works. As you mentioned, we extended existing theoretical results from prior research. However, earlier works primarily focused on physics-informed neural networks (PINNs) for predicting the solution and target function from a single fixed partial measurement of the solution. In contrast, our work extends these results to an operator learning framework, providing stability estimates for predicting the solution and target function from any given partial measurement.
> Specifically, Theorems 1 and 2 imply that, with sufficient measurements and samples, the difference between the continuous loss function $\widetilde{\mathcal{L}}(=\widetilde{\mathcal{L_{physics}}}+\widetilde{\mathcal{L_{data}}})$ and the loss function $\mathcal{L}$ in Section 2 is small. Building on this, Theorem 3 guarantees that PI-DION can accurately approximate both the solution and target function from any given partial measurement by minimizing $\mathcal{L}$. Finally, Proposition 1 establishes that, for any $\epsilon>0$, a PI-DION structure can be constructed to optimize $\mathcal{L}$ below $\epsilon$.
> In summary, minimizing the loss function $\mathcal{L}$ in Section 2 allows PI-DION to efficiently and accurately predict both solution and target function. Since this approach can be applied to any equation with stability estimates for a single measurement, our work establishes the first general framework for physics-informed operator learning in inverse problems.
>
> 6. Thank you for pointing out the possible confusion in the definition of $u$. We have included the definition of $u_l^{i}$ immediately after defining $\mathcal{L}_{data}$ in line 155. Additionally, we have corrected the input of $f$ to be two-dimensional, specified by the points $(x, y)$. These revisions have improved the clarity of our paper, and we appreciate your feedback.
>
>
> [1] Michael Prasthofer, Tim De Ryck, and Siddhartha Mishra. Variable-input deep operator networks. 2022.
>
> [2] Sung Woong Cho, Jae Yong Lee, and Hyung Ju Hwang. Learning time-dependent pde via graph neural networks and deep operator network for robust accuracy on irregular grids. 2024

---

> ### Author Response · Authors · 2024-11-15
> **Continued**
>
> Response to Questions:
>
> 1. As you mentioned, FNO and DeepONet are typically applied to forward problems. In our experiments, both models were trained in a fully supervised setting to learn the inverse mapping $u|_{\Omega_m} \rightarrow f$. PI-DIONs, on the other hand, were trained in both supervised and unsupervised settings for comparison.
>
> 2. Thank you for highlighting the loss function with labeled training data. We have now clearly defined the loss function for supervised PI-DIONs at the beginning of Section 4 to address this concern. We appreciate your valuable feedback, which has helped improve the clarity and presentation of our work.

---

> > ### Comment · Reviewer_3Fuq · 2024-11-25
> > **Feedback from reviewer 3Fuq**
> >
> > Thank you very much for your detailed responses. The responses have addressed my questions on loss functions, theoretical novelty and some training details. I still have the following concerns.
> > 1. The current architecture is a variant of DeepONet and lacks the flexibility to deal with sensor data with varying number and locations. This issue is of crucial importance for inverse problems, since in practical scenarios it is unreasonable to fix the number and locations of sensors in advance.
> > 2.  For the comparison with PINNs, the training times of PINNs in table 4 are as least 2 hours. In my own experiences, for the Reaction Diffusion equation, PINN takes much shorter time to converge on a 3090 GPU for a single instance. How many epochs did you use? And how many samples out of 1,000 did you use to get the accuracy of PINNs?
> > 3. Training 1,000 samples involves 2.000 loss terms, and you did not mention stochastic training using batches. Have you ever encountered convergence failure during training due to too many loss terms? You mentioned convergence issues in your future work, does it merely mean improving convergence rate?
> > 4. In comparison with your method PI-DIONs, PINN is flexible on the number and locations of sensor data and does not need training samples. Some meta-learning approaches have been proposed to address the retraining issue, e.g. [1].  PI-DION needs a lot of training samples and the accuracy is lower than PINN (from table 1 and table 4, the unsupervised case of PI-DION is much less accurate than PINN). Moreover, it is hard for PI-DION to generalize to samples that are far different from training samples. Could you please give more advantages of your method over PINNs besides the inference speed?
> >
> > [1] Maryam Toloubidokhti, Yubo Ye, Ryan Missel, Xiajun Jiang, Nilesh Kumar, Ruby Shrestha, and Linwei Wang. Dats: Difficulty-aware task sampler for meta-learning physics-informed neural networks. In The Twelfth International Conference on Learning Representations, 2024

---

> ### Author Response · Authors · 2024-11-25
> **Follow-up on our response to your comments**
>
> We hope this message finds you well. We would like to kindly remind you that we have submitted our response to your review comments. As the discussion period will end in two days, we would greatly appreciate it if you could let us know whether our response has addressed your concerns.
>
> If you have any additional feedback, we would be most grateful to receive it. Again, Thank you for your valuable comments and suggestions, which have been instrumental in improving our work.

---

> ### Author Response · Authors · 2024-11-25
> **Response to reviewer 3Fuq**
>
> We sincerely appreciate your time and effort in thoroughly reviewing our rebuttal response. We are grateful for this second round of discussion and have made every effort to respond promptly, as the discussion period is nearing its deadline. For detailed explanations, please refer to each of our responses.
>
> **Q. The current architecture is a variant of DeepONet and lacks the flexibility to deal with sensor data with varying number and locations. This issue is of crucial importance for inverse problems, since in practical scenarios it is unreasonable to fix the number and locations of sensors in advance.**
>
> A. Thank you for your insightful comment regarding sensor data with varying numbers and locations. We agree that addressing this issue is crucial for extending the practical applicability of the proposed methodology. In the discussion section, we outline two variants of DeepONet that are designed to handle such variability, offering a potential direction for future work.
> While this paper focuses on sensor data with fixed numbers and locations, PI-DIONs are the first models capable of rapidly approximating solutions and target functions for individual measurements. Once trained, PI-DIONs do not require retraining for newly provided measurements. Specifically tailored for function-to-function mapping, PI-DIONs can also be trained without direct supervision on the target function. To the best of our knowledge, no existing architecture aligns with this approach. To conclude,  we believe that integrating ideas from recently proposed variants of DeepONets ([1,2]) could effectively address the concern you have raised.
>
> [1] Michael Prasthofer, Tim De Ryck, and Siddhartha Mishra. Variable-input deep operator networks. 2022.
>
> [2] Sung Woong Cho, Jae Yong Lee, and Hyung Ju Hwang. Learning time-dependent pde via graph neural networks and deep operator network for robust accuracy on irregular grids. 2024
>
> **Q. For the comparison with PINNs, the training times of PINNs in table 4 are as least 2 hours. In my own experiences, for the Reaction Diffusion equation, PINN takes much shorter time to converge on a 3090 GPU for a single instance. How many epochs did you use? And how many samples out of 1,000 did you use to get the accuracy of PINNs?**
>
> A. We sincerely appreciate the time and effort you invested in reviewing our manuscript. As you pointed out, there was an error in Table 5. This mistake arose from the selection of $\lambda$, where the values of $\lambda$ for PINN and PI-DION were different for the reaction-diffusion equation. Specifically, we initially reported the training time of PINN using $(\lambda_1, \lambda_2) = (1, 100)$ with 1e+7 epochs, which was intended for PI-DION. We have now corrected Table 5 to accurately reflect the results, including the number of epochs (1e+6). To evaluate the accuracy of PINNs, we selected a single sample from our dataset. For the reaction-diffusion equation, we just tried several random samples, and the results remained consistent. Once again, we appreciate your feedback in identifying this error.
>
>
> **Q. Training 1,000 samples involves 2.000 loss terms, and you did not mention stochastic training using batches. Have you ever encountered convergence failure during training due to too many loss terms? You mentioned convergence issues in your future work, does it merely mean improving convergence rate?**
>
> A. We appreciate your insightful comment. Regarding stochastic training, we have added the following sentence in Appendix B:
>
> **All experiments were conducted on a single RTX 3090 GPU, with the batch size determined based on available memory. For the three experiments, we used either 1,000 or 500 samples per batch**
>
> While we are aware of the typical convergence failure scenarios in PINNs, such phenomena were not observed in our experiments. However, we did note that training was slow, despite the loss function steadily decreasing. This observation led us to highlight accelerating convergence as a potential area for future research. We have now revised the discussion section to make this point clearer.

---

> ### Author Response · Authors · 2024-11-25
> **Continued**
>
> **Q. In comparison with your method PI-DIONs, PINN is flexible on the number and locations of sensor data and does not need training samples. Some meta-learning approaches have been proposed to address the retraining issue, e.g. [1]. PI-DION needs a lot of training samples and the accuracy is lower than PINN (from table 1 and table 4, the unsupervised case of PI-DION is much less accurate than PINN). Moreover, it is hard for PI-DION to generalize to samples that are far different from training samples. Could you please give more advantages of your method over PINNs besides the inference speed?**
>
> A. We are truly grateful for your valuable comment. As you kindly provided, there are some meta-learning approaches to overcome the retraining issue, however, [1] only focuses on the parametric PDE, i.e., multiple forward problems. As a response to your concern, we want to emphasize that our focus is on developing a physics-informed operator network for solving inverse problems that can be trained in a **fully unsupervised manner**, with **immediate inference** once trained, and **theoretical convergence guarantees**. This is the major differentiation of PI-DIONs from such PINN algorithms.
>
> As a minor difference, the unknowns are functions in our case, while [1] considered the scalar variable $\lambda$. Additionally, they also assumed a certain distribution for $\lambda$, which may result in an inaccurate solution for an out-of-distribution sample.
>
> [1] Maryam Toloubidokhti, Yubo Ye, Ryan Missel, Xiajun Jiang, Nilesh Kumar, Ruby Shrestha, and Linwei Wang. Dats: Difficulty-aware task sampler for meta-learning physics-informed neural networks. In The Twelfth International Conference on Learning Representations, 2024
>
>
> We hope that your concerns have been properly addressed and look forward to further discussion. Thank you once again.

---

> > ### Author Response · Authors · 2024-11-26
> > **Further response to reviewer 3Fuq regarding additional experiments**
> >
> > We believe that the incorporation of variable measurement points into PI-DIONs is a significant and valuable extension as you mentioned. To address this, we conducted additional experiments adopting the architecture of variable-input deep operator networks [1]. These experiments were performed on a new dataset with irregularly distributed measurement points. The results showed a comparable relative error (about 3.83%) to the vanilla PI-DIONs, demonstrating that PI-DIONs can be effectively generalized to handle cases with irregular measurement points. We also think the error can be further reduced with careful fine-tuning.
> >
> > The results and architecture are summarized in Appendix C.3. We’ve highlighted the changes in blue.
> >
> > We are grateful for the valuable discussion, which has further improved the manuscript. Please feel free to reach out if you have any questions.
> >
> >
> > [1] Variable-input Deep Operator Networks. Michael Prasthofer, Tim De Ryck, Siddhartha Mishra. 2022.

---

> > > ### Comment · Reviewer_3Fuq · 2024-11-30
> > > **Reply to further response to reviewer 3Fuq regarding additional experiments**
> > >
> > > Thank you for your feedback on  additional experiments for irregular measurement input. Now, the proposed method is able to dealing with irregular sensor input. Accordingly, I have upgraded my score to 5.

---

> > > > ### Author Response · Authors · 2024-11-30
> > > > **Reply to reviewer 3Fuq**
> > > >
> > > > We appreciate your timely response and updated evaluation. The discussion has been incredibly helpful.

---

> ### Author Response · Authors · 2024-11-30
> **Request for Further Discussion**
>
> We hope this message finds you well. We are writing to thank you for your valuable feedback on our manuscript. Your insights have been incredibly helpful, and we deeply appreciate the time and effort you have dedicated to reviewing our work.
>
> As the discussion period is ending soon, (December 2) we are awaiting your feedback on our revised manuscript. We would also like to know if the additional experiment on the variable input case helps address your concerns. If possible, we would greatly appreciate the opportunity to have a further discussion with you to address any remaining concerns.
>
> Thank you once again for your guidance and support. we look forward to your response.

---

### Official Review · Reviewer_Y92e · 2024-11-04

**Soundness:** 2
**Presentation:** 3
**Contribution:** 2
**Rating:** 6
**Confidence:** 3

**Summary:**

The authors propose Physics-Informed Deep Inverse Operator Networks (PI-DIONs), a novel architecture for solving PDE inverse problems without requiring labeled data. Theoretically, the authors extend stability estimates from traditional inverse problem theory to the operator learning setting, and prove universal approximation theorems for PI-DIONs. Empirically, the authors validate their proposed approach through experiments on reaction-diffusion equations, Helmholtz equations, and Darcy flow, achieving SOTA performance.

**Strengths:**

- The paper engages with an important problem in SciML, learning to solve inverse problems based on physics losses without additional training data.
- The theoretical results are quite interesting. The authors extend standard stability estimates for inverse problems to the operator learning setting. Promisingly, the theorems apply to the reaction-diffusion equation and the Helmholtz equation, two standard benchmarks in the literature.
- The proposed method is simple and presented clearly and generally.
- The empirical results are promising. On three standard benchmarks, the authors demonstrate SOTA performance of supervised learning and near-SOTA of unsupervised learning, compared to supervised DeepONet and FNO.

**Weaknesses:**

- The main weakness of the paper is that the empirical results, although promising, are relatively limited and could benefit from some clarification:
  - In Table 1, PI-DION in the supervised learning setting (with 1k training examples) is shown to outperform two different DeepONets and FNOs. However, it's a bit unclear from the paper why this is true, and additional clarification about this would be helpful. Is there a difference in the model architecture / training objective / optimizer between the DeepONets and PI-DION in the supervised setting?
  - See questions for more.

**Questions:**

- Questions about experimental results:
  - What are the number of parameters of each of the models in Table 1?
  - Could the authors provide a sensitivity analysis showing how performance changes as the relative weighting between physics and data losses is varied? This would provide valuable insight into the method's robustness.
  - How does PI-DION compare to other methods for solving inverse problems, e.g. Neural Inverse Operators [1]?
  - Any explanation about why the performance hit between supervised and unsupervised PI-DION is larger for Darcy Flow and Helmholtz equation than for reaction-diffusion?

- How limiting is the assumption that there exists stability estimates for the inverse problem?
- How well do the theoretical bounds from Theorems 2, 3 match the empirical results of Table 1 (reaction-diffusion and Helmholtz)?

1. Neural Inverse Operators for Solving PDE Inverse Problems

---

> ### Author Response · Authors · 2024-11-15
> **Response to reviewer Y92e**
>
> Thank you for clearly summarizing our contributions and highlighting the strengths of our work. We completely agree with the strengths you have outlined.
>
> Response to Weaknesses:
>
> 1. The key distinction of PI-DION lies in its loss function, $\mathcal{L_{physics}}$ Directly imposing $\mathcal{L_{physics}}$ ​ on models like DeepONet and FNO is not feasible, as they do not parameterize the solution and target function as functions of $x$. To address this limitation, we developed a novel architecture that enables the incorporation of $\mathcal{L_{physics}}$  into the training process. We believe this innovative architecture and loss function are critical factors contributing to the observed performance improvements.
>
> Response to Questions:
>
> 1. Thank you for pointing out the missing information. We have addressed this by adding Table 3 in Appendix B, which provides the number of trainable parameters for all models listed in Table 1.
>
> 2. Thank you for your insightful feedback. In response to your suggestion, we performed experiments by varying the relative weights between the physics and data losses across seven configurations: $\(\lambda_1, \lambda_2) =  (100, 1), (10, 1), (1, 1), (1,0.1), (0.1, 1), (1, 10),  (1, 100)$ . As shown in Table 4 (Appendix C.2), the relative test error decreases as the weight assigned to the data loss increases. This observation aligns with our intuition that, during the early stages of training, a large $\mathcal{L_{data}}$ will push $s_{\zeta, \theta}$ in the wrong direction, as $u_{\eta, \theta}$ differs from the true solution, leading to an inaccurate $\mathcal{L_{physics}}$. We have included this discussion at the beginning of Section 4.
>
> 3. Thank you for your valuable feedback regarding the comparison with other baselines, particularly the widely used NIO. While NIO is a promising approach, a direct comparison with our model is challenging because NIO relies on fully supervised learning for operator-to-function mapping. In contrast, PI-DION is specifically designed for function-to-function mapping and can be trained without direct supervision on the target function. To the best of our knowledge, no existing architecture aligns with this approach. Therefore, we compare our model with foundational models such as FNO and DeepONet. We appreciate your understanding of this context.
>
> 4. Thank you for your critical question regarding the performance gap across problems. We believe the primary factor is the dimensionality of the problem. For the reaction-diffusion equation, the input function (partial measurement of the solution) is defined on a one-dimensional domain ($\partial\Omega$), and the target function is also defined on a one-dimensional domain ($\\{T\\}\times\Omega$). In contrast, the other two problems involve functions defined on two-dimensional domains.
> As stated in Theorem 1 and 2, higher dimensionality increases the number of samples required to approximate the continuous version of loss functions $\widetilde{\mathcal{L_{physics}}}$ and $\widetilde{\mathcal{L_{data}}}$ with their discrete versions $\mathcal{L_{physics}}$ and $\mathcal{L_{data}}$.  This is due to the growth of constants $N_{\mathcal{N}}, N_{\mathcal{B}}$ and $N$ with dimensionality, given fixed bounds $R_{\mathcal{N}}, R_{\mathcal{B}}$ and $R$ in PI-DION.
> Additionally, empirical evidence indicates that solving elliptic PDEs is inherently more challenging than parabolic PDEs, which may contribute to the observed performance gap.
>
> 5. Thank you for considering the detailed stability estimates. As mentioned in Remark 1, these estimates are valid for certain equations, such as the inverse source problem for the reaction-diffusion equation and the Helmholtz equation. However, for the permeability function in the Darcy flow equation, relevant stability estimates have not yet been established. Despite this, empirical evidence presented in Section 4 confirms that PI-DION can still be consistently applied to the Darcy flow problem with the loss function described in Lines 152 and 352. These results suggest that PI-DION shows promise for approximating various inverse operators, even in the absence of formal stability estimates.

---

> ### Author Response · Authors · 2024-11-15
> **Continued**
>
> 6. Thank you for the valuable feedback regarding the verification of the theoretical bounds presented in Theorems 2 and 3. These theorems provide a rigorous foundation for PI-DION, ensuring that the approximated solution and target function converge to the true solution and target as the loss function $\mathcal{L}$ is computed over a sufficiently large number of samples. As demonstrated in Table 5 of Appendix C.2, the relative error decreases with an increasing sample size. Since the theorems hold with a certain probability, empirically verifying the convergence rate (i.e., the order of error on sample size) would require extensive experimentation, which represents an interesting direction for future research. We have included a comment on this in the discussion section.
>
> [1] Roberto Molinaro, Yunan Yang, Bj¨orn Engquist, and Siddhartha Mishra. Neural inverse operators for solving pde inverse problems. 2023.

---

> > ### Comment · Reviewer_Y92e · 2024-11-21
> > **Thanks for the response**
> >
> > Thanks to the authors for the detailed response and additional experimental results! I have raised my score accordingly.

---

> > > ### Author Response · Authors · 2024-11-21
> > > **Thank you**
> > >
> > > We sincerely appreciate your thoughtful consideration. Please feel free to reach out if you have any further questions or would like to discuss anything further.

---

### Author Response · Authors · 2024-11-15
**General response to reviewers**

We sincerely thank the reviewers for their thoughtful and constructive feedback on our manuscript. The comments were both insightful and helpful, enabling us to make significant improvements. In our revisions, we have clarified the notation and explanations, as well as added further details, including the number of parameters, the relative weights, and the training and inference times, which are now provided in Appendices B and C. Furthermore, we have conducted additional experiments, detailed in Appendix C, to highlight and emphasize our contributions. For more specific updates, please refer to the individual responses.

We believe that most of the weaknesses and questions have been thoroughly addressed, and we look forward to further discussions.

---

### Author Response · Authors · 2024-11-26
**Response to reviewers regarding additional experiments**

We sincerely appreciate your concerns regarding the variable measurement case. To address the issue, we conducted additional experiments adopting the architecture of variable-input deep operator networks [1]. The results showed a comparable relative error (about 3.83%) to the vanilla PI-DIONs, demonstrating that PI-DIONs can be effectively generalized to handle cases with irregular measurement points. We also think the error can be further reduced with careful fine-tuning.

The results and architecture are summarized in Appendix C.3. We’ve highlighted the changes in blue.

We are grateful for the valuable discussion, which has further improved the manuscript. Please feel free to reach out if you have any questions.


[1] Variable-input Deep Operator Networks. Michael Prasthofer, Tim De Ryck, Siddhartha Mishra. 2022.

---

### Author Response · Authors · 2024-12-03
**Appreciation to reviewers**

As we conclude this discussion, we would like to express our sincere gratitude to the reviewers for their time and effort in evaluating our paper. Your valuable feedback has been immensely helpful, guiding us to make significant improvements.

The key revisions made in response to the reviewers' feedback are as follows:

1. We have **revised the loss function** to incorporate the weights $\lambda$'s and conducted various experiments to illustrate effective methods for selecting appropriate $\lambda$'s.
2. We have carried out additional experiments, including **sensativity analysis** and **ablation studies** as recommended by the reviewers.
3. We proposed a simple yet effective variation of PI-DION, named **variable-input PI-DIONs**, to handle the irregular measurement case, as suggested by the reviewers. The experimental results demonstrate that variable-input PI-DIONs can effectively handle the irregular measurement data.

We believe that the revised manuscript offers a meaningful contribution to the ICLR community and sincerely hope it meets the high standards expected by the community.

---

### Meta-Review · Area_Chair_f5h7 · 2024-12-20

**Metareview:**

The work proposes a new DeepONet based architecture for learning the solution operator of PDE-based inverse problems. Stability estimates are established and the architecture is evaluated on several benchmark problems, showing dominating performance.

**Additional Comments On Reviewer Discussion:**

While I agree with some of the reviewers that the architectural design does not contain much novelty and that the inverse problems considered are quite simple, I think the formulation and stability estimates as well as the dominating performance of the method over baselines merits publication of this paper. The authors have carried out numerous new ablations and introduced a way of handling irregularly gridded data into their methodology. I'd suggest the authors considering adding a more challenging example, but, even without it, I think establishes some very impressive numerical results.

---

### Decision · Program_Chairs · 2025-01-22

Accept (Poster)